# Transcoders Find Interpretable LLM Feature Circuits

**Jacob Dunefsky**[*]
Yale University
New Haven, CT 06511
jacob.dunefsky@yale.edu

**Philippe Chlenski**[*]
Columbia University
New York, NY 10027
pac@cs.columbia.edu

**Neel Nanda**

## Abstract

A key goal in mechanistic interpretability is circuit analysis: finding sparse sub-graphs of models corresponding to specific behaviors or capabilities. However, MLP sublayers make fine-grained circuit analysis on transformer-based language models difficult. In particular, interpretable features—such as those found by sparse autoencoders (SAEs)—are typically linear combinations of extremely many neurons, each with its own nonlinearity to account for. Circuit analysis in this setting thus either yields intractably large circuits or fails to disentangle local and global behavior. To address this we explore **transcoders**, which seek to faithfully approximate a densely activating MLP layer with a wider, sparsely-activating MLP layer. We introduce a novel method for using transcoders to perform weights-based circuit analysis through MLP sublayers. The resulting circuits neatly factorize into input-dependent and input-invariant terms. We then successfully train transcoders on language models with 120M, 410M, and 1.4B parameters, and find them to perform at least on par with SAEs in terms of sparsity, faithfulness, and human-interpretability. Finally, we apply transcoders to reverse-engineer unknown circuits in the model, and we obtain novel insights regarding the "greater-than circuit" in GPT2-small. Our results suggest that transcoders can prove effective in decomposing model computations involving MLPs into interpretable circuits. Code is available at `https://github.com/jacobdunefsky/transcoder_circuits/`.

## 1 Introduction

In recent years, transformer-based large language models (LLMs) have displayed outstanding performance on a wide variety of tasks [8, 43, 46]. However, the mechanisms by which LLMs perform these tasks are opaque by default [10, 33]. The field of mechanistic interpretablity [9] seeks to understand these mechanisms, and doing so relies on decomposing a model into **circuits** [41]: interpretable subcomputations responsible for specific model behaviors [15, 32, 42, 50].

A core problem in fine-grained circuit analysis is incorporating MLP sublayers [32, 38]. Attempting to analyze MLP neurons directly suffers from "polysemanticity" [3, 16, 24, 40]: the tendency of neurons to activate on many unrelated concepts. To address this, **sparse autoencoders (SAEs)** [7, 12, 51] have been used to perform fine-grained circuit analysis by instead looking at **features**—vectors in the model's representation space—instead of individual neurons [14, 34]. However, while SAE features are often interpretable, these vectors tend to be dense linear combinations of many neurons [36]. Thus, mechanistically understanding how an SAE feature before one or more MLP layers affects a later SAE feature may require considering an infeasible number of neurons and their nonlinearities. Prior attempts to circumvent this [14, 34] use a mix of causal interventions and gradient-based

---

[*]Equal contribution.

38th Conference on Neural Information Processing Systems (NeurIPS 2024).

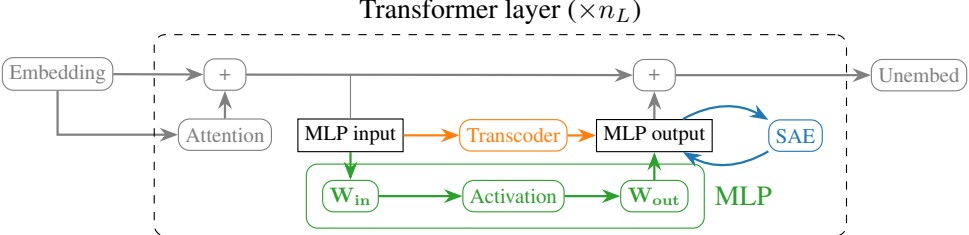

Figure 1: A comparison between **SAEs**, **MLP transcoders**, and **MLP sublayers** for a transformer-based language model. SAEs learn to reconstruct model activations, whereas transcoders imitate sublayers' input-output behavior.

approximations to MLP layers. But these approaches fail to exhibit **input-invariance**: the connections between features can only ever be described *for a given input*, and not for the model as a whole. Attempts to address this, e.g. by averaging results over many inputs, conversely lose their ability to yield **input-dependent** information that describes a connection's importance on a single input. This means that SAEs *cannot tell us about the general input-output behavior of an MLP across all inputs*.

To address why input-invariance is desirable, consider the following example: say that one has a post-MLP SAE feature and wants to see how it is computed from pre-MLP SAE features. Doing e.g. patching on one input shows that a pre-MLP feature for Polish last names is important for causing the post-MLP feature to activate. But on other inputs, would features other than the Polish last name feature also cause the post-MLP feature to fire (e.g. an English last names feature)? Could there be other inputs where the Polish last names feature fires but the post-MLP feature does not? We can see that without input-invariance, it is difficult to make general claims about model behavior.

Motivated by this, in this work, we explore **transcoders** (an idea proposed, but not explored, in Templeton et al. [47] and Li et al. [31]): wide, sparsely-activating approximations of a model's original MLP sublayer. Specifically, MLP transcoders are wide ReLU MLPs with one hidden layer that are trained to faithfully approximate the original narrower MLP sublayer's output, with an L1 regularization penalty on neuron activations to encourage sparse activations. **Our primary motivation** is to enable input-invariant feature-level circuit analysis through MLP sublayers, which allows us to understand and interpret the general behavior of circuits involving MLP sublayers.

**Our contributions.** Our main contributions are (1) to introduce a method for circuit analysis using transcoders, (2) to confirm that transcoders are a faithful and interpretable approximation to MLP sublayers, and (3) to demonstrate the utility of our circuit analysis method on detailed case studies.

After describing the architecture of transcoders in §3.1, we demonstrate in §3.2 that transcoders additionally enable circuit-finding techniques that are not possible using SAEs, and introduce a novel method for performing circuit analysis with transcoders and demonstrate that transcoders cleanly factorize circuits into *input-invariant* and *input-dependent* components.

Then, in §4, we evaluate transcoders' interpretability, sparsity, and faithfulness to the original model. Because SAEs are the standard method for finding sparse decompositions of model activations, we compare transcoders to SAEs on models up to 1.4 billion parameters and verify that transcoders are on par with SAEs or better with respect to these properties.

We apply transcoder circuit analysis to a variety of tasks in §5.1 and §5.2, including "blind case studies," which demonstrate how this approach allows us to understand features without looking at specific examples, and an in-depth analysis of the GPT2-small "greater-than circuit" previously studied by Hanna et al. [26].

## 2 Transformers preliminaries

Following Elhage et al. [15], we represent the computation of a transformer model as follows. First, the model maps input tokens (and their positions) to embeddings $\mathbf{x}_{\mathbf{pre}}^{(\mathbf{0,t})} \in \mathbb{R}^{d_{\mathrm{model}}}$, where $t$ is the token index and $d_{\mathrm{model}}$ is the *model dimensionality*. Then, the model applies a series of "layers," which map the hidden state at the end of the previous block to the new hidden state. This can be expressed as:

$$\mathbf{x}_{\mathbf{mid}}^{(\mathbf{l,t})} = \mathbf{x}_{\mathbf{pre}}^{(\mathbf{l,t})} + \sum_{\text{head } h} \mathrm{attn}^{(l,h)}\left(\mathbf{x}_{\mathbf{pre}}^{(\mathbf{l,t})}; \mathbf{x}_{\mathbf{pre}}^{(\mathbf{l,1:t})}\right) \tag{1}$$

$$\mathbf{x}_{\mathbf{pre}}^{(\mathbf{l+1,t})} = \mathbf{x}_{\mathbf{mid}}^{(\mathbf{l,t})} + \mathrm{MLP}^{(l)}\left(\mathbf{x}_{\mathbf{mid}}^{(\mathbf{l,t})}\right) \tag{2}$$

where $l$ is the layer index, $t$ is the token index, $\mathrm{attn}^{(l,h)}(\mathbf{x}_{\mathbf{pre}}^{(\mathbf{l,t})}; \mathbf{x}_{\mathbf{pre}}^{(\mathbf{l,1:t})})$ denotes the output of attention head $h$ at layer $l$ given all preceding source tokens $\mathbf{x}_{\mathbf{pre}}^{(\mathbf{l,1:t})}$ and destination token $\mathbf{x}_{\mathbf{pre}}^{(\mathbf{l,t})}$, and $\mathrm{MLP}^{(l)}(\mathbf{x}_{\mathbf{mid}}^{(\mathbf{l,t})})$ denotes the output of the layer $l$ MLP.[2]

Equation 1 shows how the **attention sublayer** updates the hidden state at token $t$, and Equation 2 shows how the **MLP sublayer** updates the hidden state. Importantly, each sublayer always *adds* its output to the current hidden state. As such, the hidden state always can be additively decomposed into the outputs of all previous sublayers. This motivates Elhage et al. [15] to refer to each token's hidden state as its **residual stream**, which is "read from" and "written to" by each sublayer.

## 3 Transcoders

### 3.1 Architecture and training

Transcoders aim to learn a "sparsified" approximation of an MLP sublayer: they approximate the output of an MLP sublayer as a sparse linear combination of feature vectors. Formally, the transcoder architecture can be expressed as

$$\mathbf{z_{TC}}(\mathbf{x}) = \mathrm{ReLU}\left(\mathbf{W_{enc}}\mathbf{x} + \mathbf{b_{enc}}\right) \tag{3}$$

$$\mathrm{TC}(\mathbf{x}) = \mathbf{W_{dec}}\mathbf{z_{TC}}(\mathbf{x}) + \mathbf{b_{dec}}, \tag{4}$$

where $\mathbf{x}$ is the input to the MLP sublayer, $\mathbf{W_{enc}} \in \mathbb{R}^{d_{\text{features}} \times d_{\text{model}}}$, $\mathbf{W_{dec}} \in \mathbb{R}^{d_{\text{model}} \times d_{\text{features}}}$, $\mathbf{b_{enc}} \in \mathbb{R}^{d_{\text{features}}}$, $\mathbf{b_{dec}} \in \mathbb{R}^{d_{\text{model}}}$, $d_{\text{features}}$ is the number of feature vectors in the transcoder, and $d_{\text{model}}$ is the dimensionality of the MLP input activations. Usually, $d_{\text{features}}$ is far greater than $d_{\text{model}}$.

Each feature in a transcoder is associated with two vectors: the $i$-th row of $\mathbf{W_{enc}}$ is the **encoder feature vector** of feature $i$, and the $i$-th column of $\mathbf{W_{dec}}$ is the **decoder feature vector** of feature $i$. The $i$-th component of $\mathbf{z_{TC}}(\mathbf{x})$ is called the **activation** of feature $i$. Intuitively, for each feature, the encoder vector is used to determine how much the feature should activate; the decoder vector is then scaled by this amount, and the resulting weighted sum of decoder vectors is the output of the transcoder. In this paper, the notation $\mathbf{f}_{\mathbf{enc}}^{(\mathbf{l,i})}$ and $\mathbf{f}_{\mathbf{dec}}^{(\mathbf{l,i})}$ is used to denote the $i$-th encoder feature vector and decoder feature vector, respectively, in the layer $l$ transcoder.

Because we want transcoders to learn to approximate an MLP sublayer's computation with a sparse linear combination of feature vectors, transcoders are trained with the following loss, where $\lambda_1$ is a hyperparameter mediating the tradeoff between sparsity and faithfulness:

$$\mathcal{L}_{TC}(\mathbf{x}) = \underbrace{\|\mathrm{MLP}(\mathbf{x}) - \mathrm{TC}(\mathbf{x})\|_2^2}_{\text{faithfulness loss}} + \underbrace{\lambda_1 \|\mathbf{z_{TC}}(\mathbf{x})\|_1}_{\text{sparsity penalty}}. \tag{5}$$

### 3.2 Circuit analysis with transcoders

We now introduce a novel method for performing feature-level circuit analysis with transcoders, which provides a scalable and interpretable way to identify which transcoder features in different layers connect to compute a given task. Importantly, this method provides insights into the general input-output behavior of MLP sublayers, which SAE-based methods cannot do.

In particular, the primary goal of circuit analysis is to identify a subgraph of the model's computational graph that is responsible for (most of) the model's behavior on a given task [11, 19, 20]; this requires a means of evaluating a computational subgraph's importance to the task in question. In order to

---

[2]Note that the "Pythia" family of models computes MLP and attention sublayer outputs in parallel. This means that Equation 2 is thus given by $\mathbf{x}_{\mathbf{pre}}^{(\mathbf{l+1,t})} = \mathbf{x}_{\mathbf{mid}}^{(\mathbf{l,t})} + \mathrm{MLP}^{(l)}\left(\mathbf{x}_{\mathbf{pre}}^{(\mathbf{l,t})}\right)$.

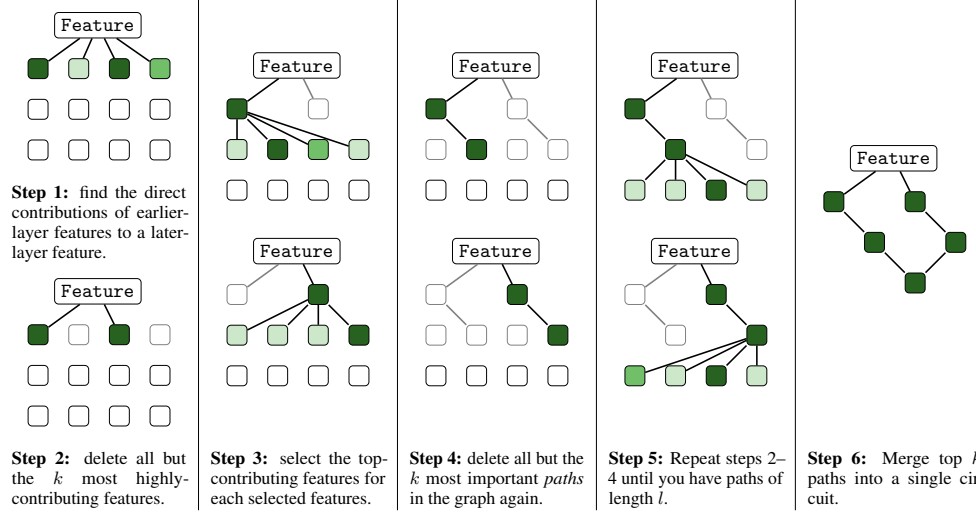

**Step 1:** find the direct contributions of earlier-layer features to a later-layer feature.

**Step 2:** delete all but the $k$ most highly-contributing features.

**Step 3:** select the top-contributing features for each selected features.

**Step 4:** delete all but the $k$ most important *paths* in the graph again.

**Step 5:** Repeat steps 2–4 until you have paths of length $l$.

**Step 6:** Merge top $k$ paths into a single circuit.

Figure 2: A visualization of the circuit-finding algorithm.

determine which edges are included in this subgraph, we thus have to compute **attributions** for each edge: how much the earlier node contributes to the later node's own contribution. Circuit analysis with SAEs thus entails computing the attribution of pre-MLP SAE features to post-MLP SAE features, *as mediated through the MLP*. Standard methods for computing attributions are causal patching (which inherently only gives information about local MLP behavior on a single input) and methods like input-times-gradient or attribution patching (which are equivalent in this setting). We will now demonstrate why these methods cannot yield information about the MLP's general behavior. Letting $\mathbf{z}$ be the activation of an earlier-layer feature, $\mathbf{z}'$ be the activation of the later-layer feature, and $\mathbf{y}$ be the activation of the MLP at layer $l'$, the input-times-gradient is given by:

$$\mathbf{z}\left(\frac{\partial \mathbf{z}'}{\partial \mathbf{z}}\right) = \mathbf{z}\left(\frac{\partial \mathbf{z}'}{\partial \mathbf{y}}\frac{\partial \mathbf{y}}{\partial \mathbf{z}}\right). \tag{6}$$

Unfortunately, not only is $\mathbf{z}$ input-dependent, but so is $\frac{\partial \mathbf{z}'}{\partial \mathbf{z}}$ as well, because $\frac{\partial \mathbf{y}}{\partial \mathbf{z}}$ is.

This means that we cannot use SAEs to understand the general behavior of MLPs on various inputs. In contrast, we will show that when we replace MLP sublayers with sufficiently faithful and interpretable transcoders, we obtain attributions that neatly factorize into *input-dependent* terms and *input-invariant* terms; the latter can be computed just from model and transcoder weights, and tell us about the MLP behavior across all inputs.

### 3.2.1 Attribution between transcoder feature pairs

We begin by showing how to compute attributions between pairs of transcoder features. This attribution is given by the product of two terms: the earlier feature's activation (which depends on the input to the model), and the dot product of the earlier feature's decoder vector with the later feature's encoder vector (which is independent of the model input).

The following is a more formal restatement. Let $z_{TC}^{(l,i)}\left(\mathbf{x}_{\mathbf{mid}}^{(\mathbf{l},\mathbf{t})}\right)$ denote the scalar activation of the $i$-th feature in the layer $l$ transcoder on token $t$, as a function of the MLP input $\mathbf{x}_{\mathbf{mid}}^{(\mathbf{l},\mathbf{t})}$ at token $t$ in layer $l$. Then for layer $l < l'$, the contribution of feature $i$ in transcoder $l$ to the activation of feature $i'$ in transcoder $l'$ on token $t$ is given by

$$\underbrace{z_{TC}^{(l,i)}\left(\mathbf{x}_{\mathbf{mid}}^{(\mathbf{l},\mathbf{t})}\right)}_{\text{input-dependent}}\underbrace{\left(\mathbf{f}_{\mathbf{dec}}^{(\mathbf{l},\mathbf{i})}\cdot\mathbf{f}_{\mathbf{enc}}^{(\mathbf{l}',\mathbf{i}')}\right)}_{\text{input-invariant}} \tag{7}$$

This expression is derived in App. D.2. Note that $\left(\mathbf{f}_{\mathbf{dec}}^{(\mathbf{l},\mathbf{i})}\cdot\mathbf{f}_{\mathbf{enc}}^{(\mathbf{l}',\mathbf{i}')}\right)$ is *input-invariant*: once the transcoders have been trained, this term does not depend on the input to the model. This term,

analyzed in isolation, can thus be viewed as providing information about the general behavior of the model. The only *input-dependent* term is $z_{TC}^{(l,i)}\left(\mathbf{x}_{\mathbf{mid}}^{(\mathbf{l,t})}\right)$, the activation of feature $i$ in the layer $l$ transcoder on token $t$. As such, this expression cleanly factorizes into a term reflecting the general input-invariant connection between the pair of features and an interpretable term reflecting the importance of the earlier feature on the current input.

### 3.2.2 Attribution through attention heads

So far, we have addressed how to find the attribution of a lower-layer transcoder feature directly on a higher-layer transcoder feature at the same token. But transcoder features can also be mediated by attention heads. We will thus extend the above analysis to account for finding the attribution of transcoder features through the OV circuit of an attention head. For a full derivation, see App. D.3.

As before, we want to understand what causes feature $i'$ in the layer $l'$ transcoder to activate on token $t$. Given attention head $h$ at layer $l$ with $l < l'$, the contribution of token $s$ at layer $l$ through head $h$ to feature $i'$ in layer $l'$ at token $t$ is given by

$$\text{score}^{(l,h)}\left(\mathbf{x}_{\mathbf{pre}}^{(\mathbf{l,t})}, \mathbf{x}_{\mathbf{pre}}^{(\mathbf{l,s})}\right)\left(\left(\left(\mathbf{W}_{\mathbf{OV}}^{(\mathbf{l,h})}\right)^T \mathbf{f}_{\mathbf{enc}}^{(l',i')}\right) \cdot \mathbf{x}_{\mathbf{pre}}^{(\mathbf{l,s})}\right), \tag{8}$$

where $\text{score}^{(l,h)}\left(\mathbf{x}_{\mathbf{pre}}^{(\mathbf{l,t})}, \mathbf{x}_{\mathbf{pre}}^{(\mathbf{l,s})}\right)$ is the attention score for head $h$ and layer $l$ from token $s$ to token $t$.

### 3.2.3 Finding computational subgraphs

Using this observation, we present a method for finding computational subgraphs. We now know how to determine, on a given input and transcoder feature $i'$, which earlier-layer transcoder features $i$ are important for causing $i'$ to activate. Once we have identified some earlier-layer features $i$ that are relevant to $i'$, then we can then recurse on $i$ to understand the most important features causing $i$ to activate by repeating this process.

Doing so iteratively (and greedily pruning all but the most important features at each step) thus yields a set of computational paths (a sequence of connected edges). These computational paths can then be combined into a computational subgraph, in such a way that each node (transcoder feature or attention head), edge, and path is assigned an attribution. A full description of the circuit-finding algorithm is presented in App. D.5. Figure 2 provides a visualization of this algorithm.

### 3.2.4 De-embeddings: a special case of input-invariant information

Earlier, we discussed how to compute the input-invariant connection between a pair of transcoder features, providing insights on general behavior of the model. A related technique is something that we call **de-embeddings**. A de-embedding vector for a transcoder feature is a vector that contains the direct effect of *the embedding of each token in the model's vocabulary* on the transcoder feature. The de-embedding vector for feature $i$ in the layer $l$ transcoder is given by $\mathbf{W_E}^T\mathbf{f}_{\mathbf{enc}}^{(\mathbf{l,i})}$, where $\mathbf{W_E}$ is the model's token embedding matrix. Importantly, this vector gives us input-invariant information about how much each possible input token would directly contribute to the feature's activation.

Given a de-embedding vector, looking at which tokens in the model's vocabulary have the highest de-embedding scores tells us about the feature's general behavior. For example, for a certain GPT2-small MLP0 transcoder feature that we investigated, the tokens with the highest scores were `oglu`, `owsky`, `zyk`, `chenko`, and `kowski`. Notice the interpretable pattern: all of these tokens come from European surnames, primarily Polish ones, suggesting that the feature generally fires on Polish surnames.

## 4 Comparison with SAEs

Transcoders were originally conceived as a variant of SAEs, and as such, there are many similarities between them. They differ only in their training objective: because SAEs are autoencoders, the faithfulness term in the SAE loss measures the reconstruction error between the SAE's output and its original input. In contrast, the faithfulness term of the transcoder loss measures the error between the transcoder's output and the original MLP sublayer's output.

Table 1: The number of interpretable features, possibly-interpretable features, and uninterpretable features for the transcoder and MLP-in SAE. Of the interpretable features, we additionally deemed 6 transcoder features, and 16 SAE features to be "context-free", meaning they appeared to fire on a single token without any evident context-dependent patterns.

|  | Transcoder | MLP-in SAE |
| --- | --- | --- |
| # interpretable | 41 | 38 |
| # maybe | 8 | 8 |
| # uninterpretable | 1 | 4 |

Because of these similarities, SAEs can be quantitatively evaluated (for sparsity and fidelity) and qualitatively evaluated (for feature interpretability) in precisely the same way as transcoders, using standard SAE evaluation methods [4, 29]. We now report the results of evaluations comparing SAEs to transcoders on these metrics, and find that transcoders are comparable to or better than SAEs.

## 4.1 Blind interpretability comparison of transcoders to SAEs

In order to evaluate the interpretability of transcoders, we manually attempted to interpret 50 random features from a Pythia-410M layer 15 transcoder and 50 random features from a Pythia-410M layer 15 SAE trained on *MLP inputs*.[3] For each feature, the examples in a subset of the OpenWebText corpus that caused the feature to activate the most were computed ahead of time. Then, the features from both the SAE and the transcoder were randomly shuffled. For each feature, the maximum-activating examples were displayed, but not whether the feature came from an SAE or transcoder. We recorded for each feature whether or not there seemed to be an interpretable pattern, and only after examining every feature did we look at which features came from where. The results, shown in Table 1, suggest transcoder features are approximately as interpretable as SAE features. This further suggests that transcoders incur no penalties compared to SAEs.

## 4.2 Quantitative comparison of transcoders to SAEs

### 4.2.1 Evaluation metrics

We evaulate transcoders *qualitatively* on their features' interpretability as judged by a human rater, and *quantitatively* on the sparsity of their activations and their fidelity to the original MLP's computation.

As a qualitative proxy measure for the interpretability of a feature, we follow Bricken et al. [7] in assuming that interpretable features should demonstrate interpretable patterns in the examples that cause them to activate. To this end, one can run the transcoder on a large dataset of text, see which dataset examples cause the feature to activate, and see if there is an interpretable pattern among these tokens. While imperfect [6], this is still a reasonable proxy for an inherently qualitative concept.

To measure the sparsity of a transcoder, one can run the transcoder on a dataset of inputs, and calculate the mean number of features active on each token (the mean $L_0$ norm of the activations). To measure the fidelity of the transcoder, one can perform the following procedure. First, run the original model on a large dataset of inputs, and measure the next-token-prediction cross entropy loss on the dataset. Then, replace the model's MLP sublayer corresponding to the transcoder *with the transcoder*, and measure the modified model's mean loss on the dataset. Now, the faithfulness of the transcoder can be quantified as the difference between the modified model's loss and the original model's loss.

### 4.2.2 Results

We trained SAEs and transcoders on activations from GPT2-small [44], Pythia-410M, and Pythia-1.4B [2]. For each model, we trained multiple SAEs and transcoders on the same inputs, but with different values of the $\lambda_1$ hyperparameter controlling the fidelity-sparsity tradeoff for each SAE and each transcoder. The transcoders were trained on MLP-in and MLP-out activations, while SAEs were

---

[3]We used SAEs trained on MLP inputs here because the interpretability case studies look at feature activations, which are solely dependent on the encoder vectors of the SAEs and transcoders. Because transcoders' encoder vectors live in MLP input space, we thought that the comparison would be most accurate if our SAEs' encoder vectors also lived in MLP input space.

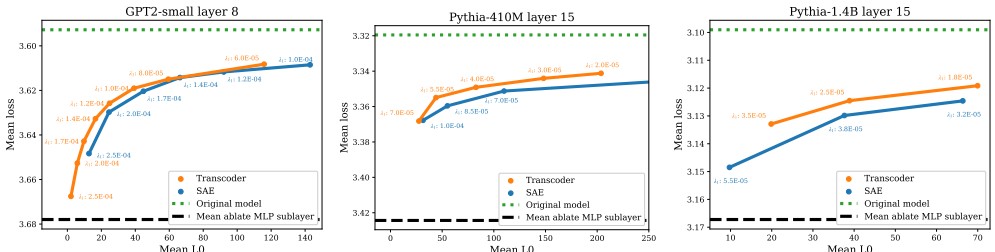

Figure 3: The sparsity-accuracy tradeoff of **transcoders** versus **SAEs** on GPT2-small, Pythia-410M, and Pythia-1.4B. Each point corresponds to a trained SAE or transcoder, and is labeled with the L1 regularization penalty $\lambda_1$ used during training.

trained on MLP-out activations (as these are the activations that MLP SAEs are typically trained on). Due to compute limitations, we used the same learning rate, which was determined via a hyperparameter sweep on *transcoders*, for both SAEs and transcoders. This means that the learning rate might not be optimal for SAEs. Nevertheless, we did perform a separate hyperparameter sweep of $\lambda_1$ for the SAEs and transcoders.

We evaluated each SAE and transcoder on the same 3.2M tokens of OpenWebText data [21]. We also recorded the loss of the unmodified and mean-ablated model (always replacing the MLP sublayer output with its mean output over the dataset) as best- and worst-case bounds, respectively.

We summarize the Pareto frontiers of the sparsity-accuracy tradeoff for all models in Figure 3. In all cases, transcoders are equal to or better than SAEs. In fact, the gap between transcoders and SAEs seems to widen on larger models. Note, however, that compute limitations prevented us from performing more exhaustive hyperparameter sweeps; as such, it might be possible that a different set of hyperparameters could have allowed SAEs to surpass transcoders. Nonetheless, these results make us optimistic that using transcoders incurs no penalties versus SAEs trained on MLP activations.

## 5 Circuit analysis case studies

### 5.1 Blind case study: reverse-engineering a feature

To understand the utility of transcoders for circuit analysis, we carried out nine **blind case studies**, where we randomly selected individual transcoder features in a ninth-layer (of 12) GPT2-small transcoder and used circuit analysis to form a hypothesis about the semantics of the feature—*without looking at the text of examples that cause the feature to activate*. In blind case studies, we use a combination of input-invariant and input-dependent information to allow us to evaluate transcoders as a tool to infer model behavior with minimal prompt information. This better reflects a key goal of mechanistic interpretability: to be able to understand model behavior on unknown, unforeseen tasks.

In contrast, reverse-engineering a feature where one already has an idea of its behavior can introduce confirmation bias. For instance, looking at activation patterns prior to circuit analysis can predispose a researcher to seek out only circuits that corroborate their interpretation of these activation patterns, potentially ignoring circuits that reveal other information about the feature. Conversely, if the circuit analysis method is faulty and yields some explanations that are not reflected in the feature activations, then the researcher might ignore those spurious explanations and thus obtain an overly-positive assessment of the circuit analysis method. The "rules of the game" for blind case studies are that:

1. The specific tokens contained in any prompt are not allowed to be directly seen. As such, prompts and tokens can only be referenced by their index in the dataset.

2. These prompts may be used to compute input-dependent information (activations and circuits), as long as the tokens themselves remain hidden.

3. Any input-invariant information, including feature de-embeddings, is allowed.

In this section, we summarise a specific blind case study, how we used our circuits to reverse-engineer feature 355 in our layer 8 transcoder. Other studies, as well as a longer description of the study summarized here, can be found in App. H.

Note that we use the following compact notation for transcoder features: `tcA[B]@C` refers to feature B in the layer A transcoder at token C.

**Building the first circuit.** We started by getting a list of indices of the top-activating prompts in the dataset for `tc8[355]`. Importantly, we did not look at the actual tokens in these prompts, as doing so would violate Rule 1. For our first input, we chose example 5701, token 37; `tc8[355]` fires at strength 11.91 on this token in this input. Our greedy algorithm for finding the most important computational paths for causing `tc8[355]@37` to fire revealed contributions from the current token (37) and earlier tokens (like 35, 36, and 31).

**Current-token features.** From token 37, we found strong contributions from `tc0[16632]@37` and `tc0[9188]@37`. Input-invariant de-embeddings of these layer 0 features revealed that they primarily activate on variants of `;`, suggesting that token 37 contributed to the feature by virtue of being a semicolon. Another feature which contributed strongly through the current token, `tc6[11831]`, showed a similar pattern. Among the top *input-invariant* connections from layer 0 transcoder features to `tc6[11831]`, we once again found the same semicolon features `tc0[16632]` and `tc0[9188]`.

**Previous-token features.** Next we checked computational paths from previous tokens through attention heads. Looking at these contextual computational paths revealed a contribution from `tc0[13196]@36`; the top de-embeddings for this feature were years like 1973, 1971, 1967, and 1966. Additionally, there was a contribution from `tc0[10109]@31`, for which the top de-embedding was (.

Furthermore, there was a contribution from `tc6[21046]@35`. The top input-invariant connections to this feature from layer 0 were `tc0[16382]` and `tc0[5468]`. The top de-embeddings for the former were tokens associated with Eastern European last names (e.g. kowski, chenko, owicz) and the top de-embeddings for the latter feature were English surnames (e.g. Burnett, Hawkins, Johnston). This heavily suggested that `tc6[21046]` was a surname feature.

Thus, the circuit revealed this pattern was important to our feature: "(-[?]-[?]-[?]-[surname]-[year]-;".

**Analysis.** We hypothesized that `tc8[355]` fires on semicolons in parenthetical citations like "(Vaswani et al. 2017; Elhage et al. 2021)". Further investigation on another input yielded a similar pattern—along with a feature whose top de-embedding tokens included Accessed, Retrieved, Neuroscience, and Springer. This bolstered our hypothesis even more.

Here, we decided to end the blind case study and check if our hypothesis was correct. Sure enough, the top activating examples included semicolons in citations such as "(Poeck, 1969; Rinn, 1984)" and "(Robinson et al., 1984; Starkstein et al., 1988)". We note that the first of these is the example at index (5701, 37) we analyzed above.

**"Restricted" blind case studies.** Because MLP0 features tend to be single-token, significant information about the original prompt can be obtained by looking at which MLP0 transcoder features are active and then taking their de-embeddings. In order to address this and more fully investigate the power of input-invariant circuit analysis, six of the eight case studies that we carried out were **restricted blind case studies**, in which all input-dependent MLP0 feature information is forbidden to use. For more details on these case studies, see Appendix H.2.

## 5.2 Analyzing the GPT2-small "greater-than" circuit

We now turn to address the "greater-than" circuit in GPT2-small previously considered by Hanna et al. [25]. They considered the following question: given a prompt such as "The war lasted from 1737 to 17", how does the model know that the predicted next year token has to be greater than 1737? In their original work, they analyzed the circuit responsible for this behavior and demonstrated that MLP10 plays an important role, looking into the operation of MLP10 at a neuronal level. We now apply transcoders and the circuit analysis tools accompanying them to this same problem.

### 5.2.1 Initial investigation

First, we used the methods from Sec. 3.2.3 to investigate a single prompt and obtain the computational paths most relevant to the task. This placed a high attribution on MLP10 features, which were in turn activated by earlier-layer features mediated by attention head 1 in layer 9. This corroborates the analysis in the original work.

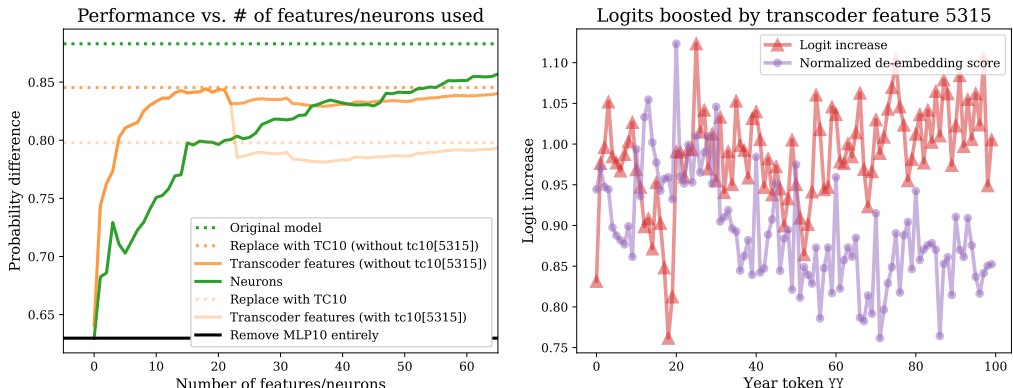

Figure 4: **Left:** Performance according to the probability difference metric when all but the top $k$ **features** or **neurons** in MLP10 are zero-ablated. **Right:** The **DLA** and **de-embedding score** for `tc10[5315]`, which contributed negatively to the transcoder's performance.

Next, we investigated which MLP10 transcoder features were most important on a variety of prompts, and how their activations are mediated by attention head 1 in layer 9. Following the original work, we generated all 100 prompts of the form "The war lasted from 17YY to 17", where YY denotes a two-digit number. We found that the MLP10 features with the highest variance in activations over this set of prompts also had top input-dependent connections from MLP0 features through attention head 1 in layer 9 whose top de-embeddings were two-digit numbers. The top *input-invariant* connections from MLP0 features through attention head 1 in layer 9 to MLP10 features *also had two-digit numbers among their top de-embedding tokens*. This positive result was somewhat unexpected, given that there are only 100 two-digit number tokens in the model's vocabulary of over 50k tokens.

We then used **direct logit attribution (DLA)** [15] to look at the effect of each transcoder feature on the predicted logits of each YY token in the model's vocabulary. These results, along with de-embedding scores for each YY token, can be seen in Figure 5. The de-embeddings scores are highest for YY tokens where years following them are boosted and years preceding them are inhibited.

### 5.2.2 Comparison with neuronal approach

Next, we compared the transcoder approach to the neuronal approach to see whether transcoders give a *sparser* description of the circuit than MLP neurons do. To do this, we computed the 100 highest-variance layer 10 transcoder features and MLP10 neurons. Then, for $1 \leq k \leq 100$, we zero-ablated all but the top $k$ features in the transcoder/neurons in MLP10 and measured how this affected the model's performance according to the **mean probability difference** metric presented in the original paper. We also evaluated the original model with respect to this metric, along with the model when MLP10 is replaced with the full transcoder.

The results are shown in the left half of Figure 4. For fewer than 24 features, the transcoder approach outperforms the neuronal approach; its performance drops sharply, however, around this point. Further investigation revealed that `tc10[5315]`, the 24th-highest-variance transcoder feature, was responsible for this drop in performance. The DLA for this feature is plotted in the right half of Figure 4. Notice how, in contrast with the three highest-variance transcoder features, `tc10[5315]` displays a flatter DLA, boosting all tokens equally. This might explain why it contributes to poor performance. To account for this, note that the left half of Figure 4 also demonstrates the performance of the transcoder when this "bad feature" is removed.

While the transcoder does not recover the full performance of the original model, it needs only a handful of features to recover most of the original model's performance; many more MLP neurons are needed to achieve the same level of performance. This suggests that the transcoder is particularly useful for obtaining a sparse, understandable approximation of MLP10. Furthermore, the transcoder features suggest a simple way that the MLP10 computation may (approximately) happen: by a small set of features that fire on years in certain ranges and boost the logits for the following years.

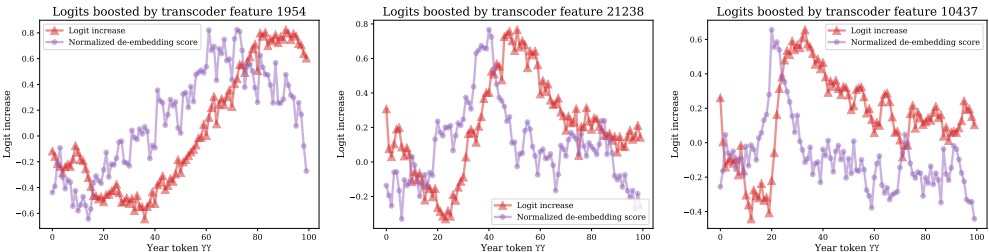

Figure 5: For the three MLP10 transcoder features with the highest activation variance over the "greater-than" dataset, and for every possible YY token, we plot the **direct logit attribution** (the extent to which the feature boosts the output probability of YY) and the **de-embedding score** (an input-invariant measurement of how much YY causes the feature to fire).

## 6 Related work

**Circuit analysis** is a common framework for exploring model internals [15, 32, 41]. A number of approaches exist to find circuits and meaningful components in models, including causal approaches [20], automated circuit discovery [11], and sparse probing [24]. Causal methods include activation patching [28, 49, 52], attribution patching [30, 39], and path patching [22, 50]. Much circuit analysis work has focused on attention head circuits [18], including copying heads [15], induction heads [42], copy suppression [35], and successor heads [23]. Methods connecting circuit analysis to SAEs include He et al. [27], Batson et al. [1] and Marks et al. [34]. Our recursive greedy circuit-finding approach was largely based on that of Dunefsky & Cohan [13].

**Sparse autoencoders** have been used to disentangle model activations into interpretable features [7, 12, 51]. The development of SAEs was motivated by the theory of superposition in neural representations [17]. Since then, much recent work has focused on exploring and interpreting SAEs, and connecting them to preexisting mechanistic interpretability techniques. Notable contributions include tools for exploring SAE features, such as SAE lens [5]; applications of SAEs to attention sublayers [29]; scaling up SAEs to Claude 3 Sonnet [48] and improved SAE architectures [45]. Transcoders have been previously proposed as a variant of SAEs under the names "predicting future activations" [47] and "MLP stretchers" [31], but not explored in detail.

## 7 Conclusion

Fine-grained circuit analysis requires an approach to handling MLP sublayers. To our knowledge, the transcoder-based circuit analysis method presented here is the only such approach *that cleanly disentangles input-invariant information from input-dependent information*. Importantly, transcoders bring these benefits without sacrificing fidelity and interpretability: when compared to state-of-the-art feature-level interpretability tools (SAEs), we find that transcoders achieve equal or better performance. We thus believe that transcoders are an improvement over other forms of feature-level interpretability tools for MLPs, such as SAEs on MLP outputs.

Future work on transcoders includes directions such as comparing the features learned by transcoders to those learned by SAEs, seeing if there are classes of features that transcoders struggle to learn, finding interesting examples of novel circuits, and scaling circuit analysis to larger models.

Overall, we believe that transcoders are an exciting new development for circuit analysis and hope that they can continue to yield deeper insights into model behaviors.

**Limitations**   Transcoders, like SAEs, are approximations to the underlying model, and the resulting error may lose key information. We find transcoders to be approximately as unfaithful to the model's computations as SAEs are (as measured by the cross-entropy loss), although we leave comparing the errors to future work. Our circuit analysis method (App. D.5) does not engage with how attention patterns are computed, and treats them as fixed. A promising direction of future work would be trying to extend transcoders to understand the computation of attention patterns, approximating the attention softmax. We only present circuit analysis results for a few qualitative case studies, and our results would be stronger with more systematic analysis.

## Impact statement

This paper seeks to advance the field of mechanistic interpretability by contributing a new tool for circuit analysis. We see this as foundational research, and expect the impact to come indirectly from future applications of circuit analysis such as understanding and debugging unexpected model behavior and controlling and steering models to be more useful to users.

## Acknowledgments and Disclosure of Funding

Jacob and Philippe were funded by a grant from AI Safety Support Ltd. Jacob was additionally funded by a grant from the Long-Term Future Fund. Philippe was additionally funded by NSF GRFP grant DGE-2036197. Compute was generously provided by Yale University.

We would like to thank Andy Arditi, Lawrence Chan, and Matt Wearden for providing detailed feedback on our manuscript. We would also like to thank Senthooran Rajamanoharan and Juan David Gil for discussions during the research process, and Joseph Bloom for advice on how to use (and extend) the SAELens library. Finally, we would like to thank Joshua Batson for a discussion that inspired us to investigate transcoders in the first place.

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

# A  Assets used

Table 2: Assets used in preparing this paper, along with licenses and links

| Asset type | Asset name | Link | License | Citation |
|------------|-----------|------|---------|----------|
| Code | TransformerLens | GitHub: TransformerLens | MIT | [37] |
| Code | SAELens | Github: SAELens | MIT | [5] |
| Data | OpenWebText | HuggingFace: OpenWebText | CC0-1.0 | [21] |
| Model | GPT2-small | HuggingFace: GPT2 | MIT | [44] |
| Model | Pythia-410M | HuggingFace: Pythia-410M | Apache-2.0 | [2] |
| Model | Pythia-1.4B | HuggingFace: Pythia-1.4B | Apache-2.0 | [2] |

# B  Compute details

The most compute-intensive parts of the research presented in this work were training the SAEs and transcoders used in Section 4.2, along with the set of GPT2-small transcoders used in Sections 5.1 and 5.2. Training all of these SAEs and transcoders involved GPUs. The SAEs and transcoders from Section 4.2 were trained on an internal cluster using an A100 GPU with 80 GB of VRAM. The VRAM used by each training run ranged from approximately 16 GB for the GPT2-small runs to approximately 60 GB for the Pythia-1.4B runs. The time taken by each training run ranged from approximately 30 minutes for the GPT2-small transcoders/SAEs to approximately 3.5 hours for the Pythia-1.4B runs.

The transcoders that were trained on each layer of GPT2-small were trained using a cloud provider, with a similar amount of time and VRAM used per training run. For these transcoders, a hyperparameter sweep was performed that involved approximately 200 training runs, which did not produce results used in the final paper.

No significant amount of storage was used, as datasets were streamed during training.

In addition to these training runs, our case studies were carried out on internal cluster nodes with GPUs. These case studies used no more than 6 GB of VRAM. The total amount of compute used during each case study is variable (depending on how in-depth one wants to investigate a case study), but is *de minimis* in comparison to the training runs. The same goes for the computation of top activating examples used in Section 4.1.

# C  SAE details

**Sparse autoencoders (SAEs)** are autoencoders trained to decompose a model's activations at a given point into a sparse linear combination of feature vectors. As a hypothetical example, given the input "Sally threw the ball to *me*", an SAE might decompose the model's activations on the token `me` into a linear combination of a "personal pronoun" feature vector, an "indirect object" feature, and a "playing sports" feature—where all of these feature vectors are automatically learned by the SAE. An SAE's architecture can be expressed as

$$\mathbf{z_{SAE}}(\mathbf{x}) = \mathrm{ReLU}\left(\mathbf{W_{enc}}\mathbf{x} + \mathbf{b_{enc}}\right) \tag{9}$$

$$\mathrm{SAE}(\mathbf{x}) = \mathbf{W_{dec}}\mathbf{z_{SAE}}(\mathbf{x}) + \mathbf{b_{dec}}, \tag{10}$$

where $\mathbf{W_{enc}} \in \mathbb{R}^{d_{\text{features}} \times d_{\text{model}}}$, $\mathbf{W_{dec}} \in \mathbb{R}^{d_{\text{model}} \times d_{\text{features}}}$, $\mathbf{b_{enc}} \in \mathbb{R}^{d_{\text{features}}}$, $\mathbf{b_{dec}} \in \mathbb{R}^{d_{\text{model}}}$, $d_{\text{features}}$ is the number of feature vectors in the SAE, and $d_{\text{model}}$ is the dimensionality of the model activations. Usually, $d_{\text{features}}$ is far greater than $d_{\text{model}}$.

Intuitively, Equation 9 transforms the neuron activations $\mathbf{x}$ into a sparse vector of SAE feature activations $\mathbf{z_{SAE}}(\mathbf{x})$. Each feature in an SAE is associated with an "encoder" vector (the $i$-th row of $\mathbf{W_{enc}}$) and a "decoder" vector (the $i$-th column of $\mathbf{W_{dec}}$). Equation 10 then reconstructs the original activations as a linear combination of decoder vectors, weighted by the feature activations.

The basic loss function on which SAEs are trained is

$$\mathcal{L}_{SAE}(\mathbf{x}) = \underbrace{\|\mathbf{x} - \mathrm{SAE}(\mathbf{x})\|_2^2}_{\text{reconstruction loss}} + \underbrace{\lambda_1 \|\mathbf{z_{SAE}}(\mathbf{x})\|_1}_{\text{sparsity penalty}}, \tag{11}$$

where $\lambda_1$ is a hyperparameter and $\| \cdot \|_1$ denotes the $L_1$ norm. The first term in the loss is the *reconstruction loss* associated with the SAE. The second term in the loss is a *sparsity penalty*, which approximately measures the number of features active on each input (the $L_1$ norm is used as a differentiable approximation of the $L_0$ "norm"). SAEs are thus pushed to reconstruct inputs accurately with a sparse number of features, with $\lambda_1$ controlling the accuracy-sparsity tradeoff. Empirically, the result of this is that SAEs learn to decompose model activations into highly interpretable features [7].

A standard method for quantitatively evaluating an SAE's performance is as follows. To measure its sparsity, evaluate the mean number of features active on any given input (the mean $L_0$). To measure its accuracy, replace the original language model's activations with the SAE's reconstructed activations and measure the change in the *language model's loss* (in this paper, this is the cross entropy loss for next token prediction).

# D    Detailed description of circuit analysis

## D.1    Notation

$\mathbf{x}_{\mathbf{pre}}^{(\mathbf{l,t})}$ denotes the hidden state for token $t$ at layer $l$ before the attention sublayer.

$\mathbf{x}_{\mathbf{mid}}^{(\mathbf{l,t})}$ denotes the hidden state for token $t$ at layer $l$ before the MLP sublayer.

When we want to refer to the hidden state of the model for all tokens, we will do by omitting the token index, writing $\mathbf{x}_{\mathbf{pre}}^{(\mathbf{l,1:t})}$ and $\mathbf{x}_{\mathbf{mid}}^{(\mathbf{l,1:t})}$. These are matrices of size $\mathbb{R}^{d_{\text{model}} \times n_{\text{tokens}}}$, where $d_{\text{model}}$ is the dimensionality of model activation vectors and $n_{\text{tokens}}$ is the number of input tokens.

The MLP sublayer at layer $l$ is denoted by $\text{MLP}^{(l)}(\cdot)$. Similarly, the transcoder for the layer $l$ MLP is denoted by $\text{TC}^{(l)}(\cdot)$.

As for attention sublayers: following Elhage et al. [15], each attention sublayer can be decomposed into the sum of $n_{\text{heads}}$ independently-acting attention heads. Each attention head depends on the hidden states of all tokens in the input, but also distinguishes the token whose hidden state is to be modified by the attention head. Thus, the output of the layer $l$ attention sublayer for token $t$ is denoted $\sum_{\text{head } h} \text{attn}^{(l,h)} \left( \mathbf{x}_{\mathbf{pre}}^{(\mathbf{l,t})}; \mathbf{x}_{\mathbf{pre}}^{(\mathbf{l,1:t})} \right)$.

Each attention head can further be decomposed as a sum over "source" tokens. In particular, the output of layer $l$ attention head $h$ for token $t$ can be written as

$$\text{attn}^{(l,h)} \left( \mathbf{x}_{\mathbf{pre}}^{(\mathbf{l,t})}; \mathbf{x}_{\mathbf{pre}}^{(\mathbf{l,1:t})} \right) = \sum_{\text{source token } s} \text{score}^{(l,h)} \left( \mathbf{x}_{\mathbf{pre}}^{(\mathbf{l,t})}, \mathbf{x}_{\mathbf{pre}}^{(\mathbf{l,s})} \right) \mathbf{W}_{\mathbf{OV}}^{(\mathbf{l,h})} \mathbf{x}_{\mathbf{pre}}^{(\mathbf{l,s})} \tag{12}$$

Here, $\text{score}^{(l,h)} : \mathbb{R}^{d_{\text{model}} \times d_{\text{model}}} \to \mathbb{R}$ is a scalar "scoring" function that weights the importance of each source token to the destination token. Additionally, $\mathbf{W}_{\mathbf{OV}}^{(\mathbf{l,h})}$ is a low-rank $\mathbb{R}^{d_{\text{model}} \times d_{\text{model}}}$ matrix that transforms the hidden state of each source token. $\text{score}^{(l,h)}$ is often referred to as the "QK circuit" of attention and $\mathbf{W}_{\mathbf{OV}}^{(\mathbf{l,h})}$ is often referred to as the "OV circuit" of attention.

## D.2    Derivation of Equation 7

We want to understand what causes feature $i'$ in the transcoder at layer $l'$ to activate on token $t$. The activation of this feature is given by

$$\text{ReLU} \left( \mathbf{f}_{\mathbf{enc}}^{(\mathbf{l',i'})} \cdot \mathbf{x}_{\mathbf{mid}}^{(\mathbf{l',t})} + b_{enc}^{(l',i')} \right), \tag{13}$$

where $\mathbf{f}_{\mathbf{enc}}^{(\mathbf{l',i'})}$ is the $i'$-th row of $\mathbf{W}_{\mathbf{enc}}$ for the layer $l'$ transcoder and $b_{enc}^{(l',i')}$ is the learned encoder bias for feature $i'$ in the layer $l'$ transcoder. Therefore, if we ignore the constant bias term $b_{enc}^{(l',i')}$, then, assuming that this feature is active (which allows us to ignore the ReLU), the activation of feature $i'$ depends solely on $\mathbf{f}_{\mathbf{enc}}^{(\mathbf{l',i'})} \cdot \mathbf{x}_{\mathbf{mid}}^{(\mathbf{l',t})}$. Because of residual connections in the transformer, $\mathbf{x}_{\mathbf{mid}}^{(\mathbf{l',t})}$ can be decomposed as the sum of the outputs of all previous components in the model. For instance, in a

two-layer model, if $\mathbf{x}_{\mathbf{mid}}^{(\mathbf{2,t})}$ is the hidden state of the model right before the second MLP sublayer, then

$$\mathbf{x}_{\mathbf{mid}}^{(\mathbf{2,t})} = \sum_h \text{attn}^{(2,h)}\left(\mathbf{x}_{\mathbf{pre}}^{(\mathbf{2,t})}; \mathbf{x}_{\mathbf{pre}}^{(\mathbf{2,1:t})}\right) + \text{MLP}^{(1)}\left(\mathbf{x}_{\mathbf{mid}}^{(\mathbf{1,t})}\right) + \sum_h \text{attn}^{(1,h)}\left(\mathbf{x}_{\mathbf{pre}}^{(\mathbf{1,t})}; \mathbf{x}_{\mathbf{pre}}^{(\mathbf{1,1:t})}\right).$$

$$(14)$$

Because of linearity, this means that the amount that $\text{MLP}^{(1)}\left(\mathbf{x}_{\mathbf{mid}}^{(\mathbf{1,t})}\right)$ contributes to $\mathbf{f}_{\mathbf{enc}}^{(\mathbf{2,i'})} \cdot \mathbf{x}_{\mathbf{mid}}^{(\mathbf{2,t})}$ is given by

$$\mathbf{f}_{\mathbf{enc}}^{(\mathbf{2,i'})} \cdot \text{MLP}^{(1)}\left(\mathbf{x}_{\mathbf{mid}}^{(\mathbf{1,t})}\right). \tag{15}$$

This is generally true for understanding the contribution of MLP $l$ to the activation of feature $i'$ in transcoder $l'$, whenever $l < l'$.

Now, if the layer $l$ transcoder is a sufficiently good approximation to the layer $l$ MLP, we can replace the latter with the former: $\mathbf{f}_{\mathbf{enc}}^{(\mathbf{l',i'})} \cdot \text{MLP}^{(l)}\left(\mathbf{x}_{\mathbf{mid}}^{(\mathbf{1,t})}\right) \approx \mathbf{f}_{\mathbf{enc}}^{(\mathbf{l',i'})} \cdot \text{TC}^{(l)}\left(\mathbf{x}_{\mathbf{mid}}^{(\mathbf{1,t})}\right)$. We can further decompose this into individual transcoder features: $\text{TC}^{(l)}\left(\mathbf{x}_{\mathbf{mid}}^{(\mathbf{1,t})}\right) = \sum_{\text{feature } j} z_{TC}^{(l,j)}(\mathbf{x}_{\mathbf{mid}}^{(\mathbf{1,t})})\mathbf{f}_{\mathbf{dec}}^{(\mathbf{1,j})}$. Thus, again taking advantage of linearity, we have

$$\mathbf{f}_{\mathbf{enc}}^{(\mathbf{l',i'})} \cdot \text{MLP}^{(l)}\left(\mathbf{x}_{\mathbf{mid}}^{(\mathbf{1,t})}\right) \approx \mathbf{f}_{\mathbf{enc}}^{(\mathbf{l',i'})} \cdot \sum_{\text{feature } j} z_{TC}^{(l,j)}(\mathbf{x}_{\mathbf{mid}}^{(\mathbf{1,t})})\mathbf{f}_{\mathbf{dec}}^{(\mathbf{1,j})} \tag{16}$$

$$= \sum_{\text{feature } j} z_{TC}^{(l,j)}(\mathbf{x}_{\mathbf{mid}}^{(\mathbf{1,t})})\left(\mathbf{f}_{\mathbf{enc}}^{(\mathbf{l',i'})} \cdot \mathbf{f}_{\mathbf{dec}}^{(\mathbf{1,j})}\right) \tag{17}$$

Therefore, the attribution of feature $i$ in transcoder $l$ on token $t$ is given by

$$z_{TC}^{(l,j)}(\mathbf{x}_{\mathbf{mid}}^{(\mathbf{1,t})})\left(\mathbf{f}_{\mathbf{enc}}^{(\mathbf{l',i'})} \cdot \mathbf{f}_{\mathbf{dec}}^{(\mathbf{1,j})}\right). \tag{18}$$

### D.3 Attribution through attention heads

So far, we have addressed how to find the attribution of a lower-layer transcoder feature directly on a higher-layer transcoder feature at the same token. But transcoder features can also be mediated by attention heads. We will thus extend the above analysis to account for finding the attribution of transcoder features through the OV circuit of an attention head.

As before, we want to understand what causes feature $i'$ in the layer $l'$ transcoder to activate on token $t$. Given attention head $h$ at layer $l$ with $l < l'$, the same arguments as before imply that the contribution of this attention head to feature $i'$ is given by $\mathbf{f}_{\mathbf{enc}}^{(\mathbf{l',i'})} \cdot \text{attn}^{(l,h)}\left(\mathbf{x}_{\mathbf{pre}}^{(\mathbf{1,t})}; \mathbf{x}_{\mathbf{pre}}^{(\mathbf{1,1:t})}\right)$. This can further be decomposed as

$$\mathbf{f}_{\mathbf{enc}}^{(\mathbf{l',i'})} \cdot \left(\sum_{\text{source token } s} \text{score}^{(l,h)}\left(\mathbf{x}_{\mathbf{pre}}^{(\mathbf{1,t})}, \mathbf{x}_{\mathbf{pre}}^{(\mathbf{1,s})}\right)\mathbf{W}_{\mathbf{OV}}^{(\mathbf{1,h})}\mathbf{x}_{\mathbf{pre}}^{(\mathbf{1,s})}\right) \tag{19}$$

$$= \sum_{\text{source token } s} \text{score}^{(l,h)}\left(\mathbf{x}_{\mathbf{pre}}^{(\mathbf{1,t})}, \mathbf{x}_{\mathbf{pre}}^{(\mathbf{1,s})}\right)\left(\left(\mathbf{f}_{\mathbf{enc}}^{(\mathbf{l',i'})}\right)^T \mathbf{W}_{\mathbf{OV}}^{(\mathbf{1,h})}\mathbf{x}_{\mathbf{pre}}^{(\mathbf{1,s})}\right) \tag{20}$$

$$= \sum_{\text{source token } s} \text{score}^{(l,h)}\left(\mathbf{x}_{\mathbf{pre}}^{(\mathbf{1,t})}, \mathbf{x}_{\mathbf{pre}}^{(\mathbf{1,s})}\right)\left(\left(\left(\mathbf{W}_{\mathbf{OV}}^{(\mathbf{1,h})}\right)^T \mathbf{f}_{\mathbf{enc}}^{(\mathbf{l',i'})}\right) \cdot \mathbf{x}_{\mathbf{pre}}^{(\mathbf{1,s})}\right). \tag{21}$$

From this, we now have that the contribution of token $s$ at layer $l$ through head $h$ is given by

$$\text{score}^{(l,h)}\left(\mathbf{x}_{\mathbf{pre}}^{(\mathbf{1,t})}, \mathbf{x}_{\mathbf{pre}}^{(\mathbf{1,s})}\right)\left(\left(\left(\mathbf{W}_{\mathbf{OV}}^{(\mathbf{1,h})}\right)^T \mathbf{f}_{\mathbf{enc}}^{(\mathbf{l',i'})}\right) \cdot \mathbf{x}_{\mathbf{pre}}^{(\mathbf{1,s})}\right). \tag{22}$$

The next step is to note that $\mathbf{x}_{\mathbf{pre}}^{(\mathbf{1,s})}$ can, in turn, be decomposed into the output of MLP sublayers (or alternatively, transcoder features), the output of attention heads, and the original token embedding. These previous-layer components affect the contribution to the original feature through both the QK circuit of attention and the OV circuit. This means that these previous-layer components can have very nonlinear effects on the contribution. We address this by following the standard practice

introduced by Elhage et al. [15], which is to treat the QK circuit scores $\text{score}^{(l,h)}\left(\mathbf{x}_{\mathbf{pre}}^{(\mathbf{l,t})}, \mathbf{x}_{\mathbf{pre}}^{(\mathbf{l,s})}\right)$ as fixed, and only look at the contributions through the OV circuit. While this does prevent us from understanding the extent to which transcoder features contribute to phenomena such as QK composition, nevertheless, the OV circuit alone is extremely informative. After all, if the QK circuit determines which tokens information is taken from, then the OV circuit determines what information is taken from each token—and this can prove immensely valuable in circuit analysis.

Thus, let us continue by treating the QK scores as fixed. Referring back to Equation 22, if $\mathbf{y}$ is the output of some previous layer component, which exists in the residual stream $\mathbf{x}_{\mathbf{pre}}^{(\mathbf{l,s})}$, then the contribution of $\mathbf{y}$ to the original transcoder feature $i'$ through the OV circuit of layer $l$ attention head $h$ is given by $\mathbf{y} \cdot \mathbf{p}'$, where

$$\mathbf{p}' = \text{score}^{(l,h)}\left(\mathbf{x}_{\mathbf{pre}}^{(\mathbf{l,t})}, \mathbf{x}_{\mathbf{pre}}^{(\mathbf{l,s})}\right)\mathbf{p}, \text{ and} \tag{23}$$

$$\mathbf{p} = \left(\mathbf{W}_{\mathbf{OV}}^{(\mathbf{l,h})}\right)^T \mathbf{f}_{\mathbf{enc}}^{(\mathbf{l',i'})}. \tag{24}$$

One way to look at this is that $\mathbf{p}'$ is a *feature vector*. Just like with transcoder features, the extent to which the feature vector $\mathbf{p}'$ is activated by a given vector $\mathbf{y}$ is given by the dot product of $\mathbf{y}$ and $\mathbf{p}'$. Treating $\mathbf{p}'$ as a feature vector like this means that *we can extend all of the techniques presented in Section 3.2 to analyze* $\mathbf{p}'$. For example, we can take the de-embedding of $\mathbf{p}'$ to determine which tokens in the model's vocabulary *when mediated by the OV circuit of layer $l$ attention head $h$* cause layer $l'$ transcoder feature $i'$ to activate the most. We can also replace the $\mathbf{f}_{\mathbf{enc}}^{(\mathbf{l',i'})}$ term in Equation 7 with $\mathbf{p}'$ in order to obtain input-invariant and input-dependent information about which transcoder features *when mediated by this OV circuit* make the greatest contribution to the activation of layer $l'$ transcoder feature $i'$. In this manner, we have extended our attribution techniques to deal with attention.

### D.4  Recursing on a single computational path

At this point, we understand how to obtain the attribution from an earlier-layer transcoder feature/attention head to a later-layer feature vector. The next step is to understand in turn what contributes to these earlier-layer features or heads. Doing so will allow us to iteratively compute attributions along an entire computational graph.

To do this, we will extend the intuition presented in Equation 23 and previously discussed by Dunefsky & Cohan [13], which is to propagate our feature vector backwards through the computational path.[4] Starting at the end of the computational path, for each node in the computational path, we compute the attribution of the node towards causing the current feature vector to activate; we then compute a new feature vector, and repeat the process using the preceding node and this new feature vector.

In particular, at every node, we want to compute the new feature vector $\mathbf{f}$ such that it satisfies the following property. Let $c'$ be a node (e.g. a transcoder feature or an attention head), $\mathbf{x}'$ be the vector of input activations to the node $c'$ (i.e. the residual stream activations before the node $c'$), $\mathbf{y}'$ be the output of $c'$, $a'$ be the attribution of $c'$ to some later-layer feature, and $\mathbf{f}'$ be the current feature vector to which we are computing the attribution of $c'$. Noting that $a' = \mathbf{f}' \cdot \mathbf{y}'$, then we want our new feature vector $\mathbf{f}$ to satisfy

$$\mathbf{f} \cdot \mathbf{x}' = a'. \tag{25}$$

This is because if $\mathbf{f}$ satisfies this property, then we can take advantage of the linearity of the residual stream to easily calculate the attribution from an earlier-layer component $c$ to the current node $c'$. In particular, if the output of $c$ is the vector $\mathbf{y}$, then this attribution is just given by $\mathbf{f} \cdot \mathbf{y}$. Another important consequence of Equation 25 and the linearity of the residual stream is that the total attribution $a'$ of node $c'$ is given by

$$a' = \sum_{\mathbf{y}} \mathbf{f} \cdot \mathbf{y} \tag{26}$$

where we sum over all the outputs $\mathbf{y}$ of all earlier nodes in the model's computational graph (including transcoder features and attention heads, but also token embeddings and learned constant bias vectors, which are leaf nodes in the computational graph).

---

[4]The similarity to backpropagation is not coincidental, as it can be shown that the method about to be described computes the "input-times-gradient" attribution often used in the explanability literature.

If $c'$ is attention head $h$ in layer $l$ and we are considering the contribution from the input activations $\mathbf{x}_{\mathbf{pre}}^{(\mathbf{l,s})}$ at source token position $s$, then Equation 22 tells us that

$$\mathbf{f} = \text{score}^{(l,h)}\left(\mathbf{x}_{\mathbf{pre}}^{(\mathbf{l,t})}, \mathbf{x}_{\mathbf{pre}}^{(\mathbf{l,s})}\right)\left(\left(\mathbf{W}_{\mathbf{OV}}^{(\mathbf{l,h})}\right)^T \mathbf{f}'\right) \quad (27)$$

where token position $t$ is the token position corresponding to the later-layer feature $\mathbf{f}'$. And if $c'$ is transcoder feature $i$ at layer $l$, then Equation 18 implies that

$$\mathbf{f} = \left(\mathbf{f}' \cdot \mathbf{f}_{\mathbf{dec}}^{(\mathbf{l,i})}\right)\mathbf{f}_{\mathbf{enc}}^{(\mathbf{l,i})}. \quad (28)$$

There is one caveat, however, that must be noted. Before every sublayer in the transformer architectures considered in this paper (that is, before every MLP sublayer and attention sublayer), there is a LayerNorm nonlinearity. Neel Nanda [39] provides intuition that LayerNorm nonlinearities can be approximated as a linear transformation that scales its input by a constant; Dunefsky & Cohan [13] provide further theoretical motivation and empirical results suggesting that this is reasonable. We follow this approach in our circuit analysis by multiplying each $\mathbf{f}$ feature vector by the appropriate LayerNorm "scaling constant" (which is empirically estimated by taking the ratio of the norm of the pre-LayerNorm activation vector to the post-LayerNorm activation vector).

### D.5   Full circuit-finding algorithm

At this point, we are ready to present the full version of our circuit-finding algorithm. The greedy computational-path-finding algorithm is presented as Algorithm 1. This algorithm incorporates the ideas presented in App. D.4 in order to evaluate the attribution of nodes in computational paths; given a set of computational paths of length $L$, it obtains a set of important computational paths of length $L + 1$ by computing all possible extensions to the current length-$L$ paths, and then keeping only the $N$ paths with the highest attributions. Note that for the purpose of clarity, the description presented here is less efficient than our actual implementation; it also does not include the LayerNorm scaling constants discussed above.

Next, given a set of computational paths, Algorithm 2 converts this set into a single computational graph. The main idea is to combine all of the paths into a single graph such that the attribution of a node in the graph is the sum of its attributions in all distinct computational paths beginning at that node. Similarly, the attribution of an edge in the graph is the sum of its attributions in all distinct computational paths beginning with that edge. This prevents double-counting of attributions. Assuming zero transcoder error, Equation 26 implies that in a graph produced by Algorithm 2 from the full set of computational paths in the model (including bias terms), the attribution of each node is the sum of the attributions of all of the incoming edges to that node. To account for transcoder error, and to account for the fact that not all computational paths are included in the graph, error nodes can be added to the graph, following the approach of Marks et al. [34].

## E   Details on Section 4.2 SAE/transcoder training

In this section, we provide details on the hyperparameters used to train the SAEs and transcoders evaluated in Section 4.2.

All SAEs and transcoders were trained with a learning rate of $2 \cdot 10^{-5}$ using the Adam optimizer. Hyperparameters (learning rate and $\lambda_1$ sparsity coefficient) were chosen largely based on trial-and-error.

The loss functions used were the vanilla SAE and transcoder loss functions as specified in Section 3.1 and Appendix C. No neuron resampling methods were used during training.

SAEs were trained on output activations of the MLP layer. Transcoders were trained on the post-LayerNorm input activations to the MLP layer and the output activations of the MLP layer. We chose to train SAEs on the output activations because when measuring cross-entropy loss with transcoders, the output activations of the MLP are replaced with the transcoder output; it is thus most valid to compare transcoders to SAEs that replace the MLP output activations as well.

The number of features in the SAEs and transcoders was always $32\times$ the dimensionality of the model on which they were trained. For GPT2-small, the model dimensionality is 768. For Pythia-410M, the model dimensionality is 1024. For Pythia-1.4B, the model dimensionality is 2048.

---

**Algorithm 1** Greedy computational-path-finding

---

**Input:**
$\mathbf{f}'$    A feature vector
$l'$    The layer from which $\mathbf{f}'$ came.
$t$    The token position associated with feature $f'$.
$a$    The activation of $\mathbf{f}'$
$L$    The number of iterations to pathfind for
$N$    The number of paths to retain after each iteration
The input prompt on which we will perform circuit analysis
**Output:**
A set of computational paths important for causing $\mathbf{f}'$ to activate
**Initialize** $\mathcal{P} \leftarrow \{[(\mathbf{f}', l', t', a')]\}$ {$\mathcal{P}$ will be our working set of computational paths. Each computational path is a list of feature vectors paired with their attributions. }
**Initialize** $\mathcal{P}_{out} \leftarrow \{\}$ {This will contain our output}
Run the model on the input prompt, caching all of its activations.
**while** $L > 0$ **do**
    **Initialize** $\mathcal{P}_{next} \leftarrow \{\}$ {This will contain the next iteration of computational paths}
    **for** each $P \in \mathcal{P}$ **do**
       Set $\mathbf{f_{cur}}, l_{cur}, t_{cur}, a_{cur}$ to the values in the last element of $P$
       **Initialize** $\mathcal{A} \leftarrow \{\}$ {The set of attributions of all lower-layer features}
       **for** each transcoder feature $i$ in layer $l$ where $l < l_{cur}$ **do**
          Insert $\left( \left(\mathbf{f_{cur}} \cdot \mathbf{f}_{\mathbf{dec}}^{(l,i)}\right) \mathbf{f}_{\mathbf{enc}}^{(l,i)}, l, t, \mathbf{z_{TC}}(\mathbf{x}_{\mathbf{mid}}^{(l,t')}) \left(\mathbf{f_{cur}} \cdot \mathbf{f}_{\mathbf{dec}}^{(l,i)}\right)\right)$ into $\mathcal{A}$
       **end for**
       **for** each attention head $h$ in layer $l$ at token $t$ where $l < l_{cur}$ and $t \le t_{cur}$ **do**
          Compute the attention score $S \leftarrow \text{score}^{(l,h)} \left( \mathbf{x}_{\mathbf{pre}}^{(\mathbf{l_{cur}}, \mathbf{t_{cur}})}, \mathbf{x}_{\mathbf{pre}}^{(\mathbf{l,t})} \right)$
          Compute the feature vector $\mathbf{f_{new}} \leftarrow S \left( \left( \mathbf{W}_{\mathbf{OV}}^{(\mathbf{l,h})} \right)^T \mathbf{f_{cur}} \right)$
          Compute the attribution $a_{new} \leftarrow \mathbf{f_{new}} \cdot \mathbf{x}_{\mathbf{pre}}^{(\mathbf{l,t})}$
          Insert $(\mathbf{f_{new}}, l, t, a_{new})$ into $\mathcal{A}$
       **end for**
       Compute the embedding attribution $a_{embed} \leftarrow \mathbf{f_{cur}} \cdot \mathbf{x}_{\mathbf{pre}}^{\mathbf{0,t_{cur}}}$
       Insert $(0, 0, t_{cur}, a_{embed})$ into $\mathcal{A}$
       **for** each $(\mathbf{f_{new}}, l_{new}, t_{new}, a_{new}) \in \mathcal{A}$ **do**
          **if** $a_{new}$ is among the top $N$ values of $a_{new}$ contained in $\mathcal{A}$ **then**
             Append $(\mathbf{f_{new}}, l_{new}, t_{new}, a_{new})$ to path $P$ and insert into $\mathcal{P}_{next}$
          **end if**
       **end for**
    **end for**
    Remove all paths in $\mathcal{P}_{next}$ except for the paths where the attribution of the earliest-layer feature vector in the path is among the top $N$ in $\mathcal{P}_{next}$
    Append all paths in $\mathcal{P}_{next}$ to $\mathcal{P}_{out}$
    $\mathcal{P} \leftarrow \mathcal{P}_{next}$
    $L \leftarrow L - 1$
**end while**
**return** $\mathcal{P}_{out}$

---

The SAEs and transcoders were trained on 60 million tokens of the OpenWebText dataset. The batch size was 4096 examples per batch. Each example contains a context window of 128 tokens; when evaluating the SAEs and transcoders, we did so on examples of length 128 tokens as well.

The same random seed (42) was used to initialize all SAEs and transcoders during the training process. In particular, this meant that training data was received in the same order by all SAEs and transcoders.

**Algorithm 2** Paths-to-graph

---

**Input:**
$\mathcal{P}$    A set of computational paths
**Output:** $\mathcal{G} = (\mathcal{V}, \mathcal{E})$ A computational graph formed from the paths of $\mathcal{P}$.
**Initialize** $\mathcal{S} \leftarrow \{\}$ {A set of already-seen computational path prefixes, to prevent us from double-counting attributions}
**Initialize** $\mathcal{V} \leftarrow \{\}$ {A dictionary mapping nodes to their attributions}
**Initialize** $\mathcal{E} \leftarrow \{\}$ {A dictionary mapping edges (node pairs) to their attributions}
**for** Each $P$ in $\mathcal{P}$ **do**
   **for** $i \in [1 \ldots |P|]$ **do**
      $s \leftarrow$ the prefix of $P$ up to and including the $i$-th element
      **if** $s \in \mathcal{S}$ **then**
         Skip this iteration of the loop.
      **end if**
      Insert $s$ into $\mathcal{S}$.
      **if** $s$ has length 1 **then**
         Let $n$ be the only node in $s$.
         Set $\mathcal{V}[n]$ to the attribution of $n$.
      **else**
         Set $n_{parent} \leftarrow P[i-1], n_{child} \leftarrow P[i]$ {Earlier-layer nodes come later in the computational paths returned by Algorithm 1}
         Add the attribution of $n_{child}$ to $\mathcal{V}[n_{child}]$
         Add the attribution of $n_{child}$ to $\mathcal{E}[(n_{child}, n_{parent})]$
      **end if**
   **end for**
**end for**
**return** $\mathcal{V}, \mathcal{E}$

---

# F    Details on Section 4.1

The transcoder used in the interpretability comparison was the Pythia-410M layer 15 transcoder trained with $\lambda_1$ sparsity coefficient $5.5 \times 10^{-5}$ from Section 4.2. The SAE used in the comparison was a Pythia-410M layer 15 SAE trained on MLP inputs with $\lambda_1 = 7.0 \times 10^{-5}$. We used an SAE trained on MLP inputs rather than one trained on MLP outputs (as in § 4.2) because the interpretability comparison involves looking at which examples cause features to activate. This, in turn, is wholly determined by the encoder feature vectors. Because the transcoder's encoder feature vectors live in the MLP input space, it is thus most valid to compare the transcoder to an SAE whose encoder feature vectors also live in the MLP input space.

This transcoder-SAE pair was chosen because the transcoder and SAE sit at very similar points on the $L_0$-cross-entropy Pareto frontier: the transcoder has an $L_0$ of 44.04 and a cross-entropy of 3.35 nats, while the SAE has an $L_0$ of 47.85 and a cross-entropy of 3.36 nats. Pythia-410M was chosen as the model with the view that its features were likely to be more interesting than those of GPT2-small, while requiring less computational power to determine top activating examples than Pythia-1.4B would. Layer 15 was chosen largely heuristically, because we believed that this layer is late enough in the model to contain complex features, while not so late in the model that features are primarily encapsulating information about which tokens come next.

In Table 1, we refer to "context-free" features that interpretable features that seemed to fire on a single token (or two tokens) regardless of the context in which they appeared. Examples of features in all four categories ("interpretable", "maybe interpretable", "uninterpretable", and "context-free"), along with the exact annotation used by the human rater, can be found in Figure 6.

# G    Details on Section 5.2

To obtain the de-embedding scores shown in Figures 5 and 4, the following method was used. First, we used the method presented in Appendix D.3 to determine which MLP0 transcoder features had the highest input-invariant connections to the given MLP10 transcoder feature through attention head 1 in

▼ Percentage in band: 0.0009%
the Manhattan Project in as little as three months, Example 17558, token 79
' and within a matter of clicks you will be Example 3561, token 49
Hotels in less than 5 years . Example 17970, token 32
polishing cloth in a matter of seconds will take the Example 13920, token 17
stop hearts within a matter of seconds or minutes . Example 22826, token 69
social anxiety… in just 30 days ! Example 24769, token 116
approximately, in a matter of minutes . This is Example 20220, token 116
However, in a matter of a day, Angl Example 9682, token 39
els in less than 5 years . Phot Example 17970, token 33

▼ Percentage in band: 0.0079%
can create one in a few seconds . Example 14303, token 116
in size in just a few years and is held Example 18275, token 60
? In only 10 days , the death Example 3118, token 74
hit the market in two years' time . It Example 11736, token 12
per cent in only 2 minutes . The battery will Example 12905, token 43
Champions League knockout stage in less than three weeks . Example 13928, token 94
out via their website within days of its launch. Example 13531, token 120
previously ring the bell within 30 seconds could now wait Example 5204, token 76
be monetized entirely in three months of POM Example 13359, token 99
, bath salts, in just a moment . Example 23976, token 96

▼ Percentage in band: 0.0190%
The video will start in 8 Cancel Play now Example 8857, token 81
whole of Africa in a year ." Image Example 18375, token 44
two withdrawals "within minutes ", he said . Example 15583, token 123
L: and sold in less than 10 mins at Example 9197, token 9
. However, in a matter of a day, Example 9682, token 38
were made in a relatively short period of time, Example 12285, token 7
80% within a period of five months . In Example 19894, token 42
functionality as possible in less than 7KB gz Example 19455, token 34
<|endoftext|> within a single monthly billing cycle, Example 9217, token 3
Tyran Primus in 745M41, Example 12100, token 121

(a) Top-activating examples for a feature annotated as "interpretable". The specific annotation was `local context feature, fires on phrases describing short amounts of time.`

▼ Percentage in band: 0.0009%
also diverted from other boring things, whether by Example 24400, token 15
thought bike lanes were too boring , so he started Example 23423, token 88
had assumed administrative work was boring, mundane and routine Example 13895, token 50
Enough chit-chat - here are the Example 12566, token 14
if you find it distracting . --- Example 22454, token 23
diverted from other boring things, whether by the Example 24400, token 16
Live: Otherwise it gets boring [24/07 Example 7489, token 73
out at you, move on . Here Example 10507, token 82

▼ Percentage in band: 0.0048%
screen. They're boring . Example 12952, token 81
But enough of that -- here's' footage Example 16040, token 40
get down to the boring stuff . C Example 13835, token 98
but also diverted from other boring things, whether Example 24400, token 14
seen as a source of distraction . A core member Example 2112, token 19
then you start to get bored . Where's the Example 2970, token 113
the rest of that season . Sound good? Okay Example 4906, token 17
negatives out of the way : I am a Example 23798, token 18
watch feels kind of staid and lame . This Example 12004, token 26
'm like, forget it ! ' Example 7965, token 23

▼ Percentage in band: 0.0240%
perhaps you are too busy and rather pat a l Example 8695, token 53
composing text, in favor of making typographical choices Example 15724, token 15
The sarcasm was lost to Ras. "T Example 25323, token 109
ough chit-chat - here are the 7 Example 12566, token 15
SLT is beyond the scope of this article, Example 1795, token 8
by shifting attention to something else , if only for Example 2594, token 75
flag]] up there for now to try and avoid Example 9816, token 12
resting and getting away from all the Starcraft Example 2789, token 73
would be to put it behind you," Dit Example 16061, token 52
verify you're not a robot by clicking the box Example 1741, token 43

(b) Top-activating examples for a feature annotated as "maybe interpretable". The specific annotation was `local context feature for boredom?  MAYBE.`

▼ Percentage in band: 0.0004%
Firstpost is now on Whats App. For the Example 644, token 87
ARES Facebook Twitter Google Whats app Pinterest Print Mail Example 16535, token 22
Tech2 is now on Whats App. For all Example 3422, token 99
, sign up for our Whats App services. Just Example 17125, token 30
, sign up for our Whats App services. Just Example 17931, token 75
SHARES Facebook Twitter Whats app Linkedin Example 10329, token 33
, sign up for our Whats App services. Just Example 3422, token 117
, Viber, and Whats App constantly chime Example 4390, token 79

▼ Percentage in band: 0.0003%
encrypted service like Signal, Whats App or Skype, Example 17898, token 106
chats through messaging app Whats App that McIn Example 6567, token 22
a 'Task Force' Whats App group while em Example 4881, token 104
labor. ADVERTISEMENT Thanks for watching! Visit Example 23004, token 118
intersections. ADVERTISEMENT Thanks for watching! Visit Example 18979, token 74
000. ADVERTISEMENT Thanks for watching! Visit Example 4368, token 63
well. ADVERTISEMENT Thanks for watching! Visit Example 11981, token 100

▼ Percentage in band: 0.0003%
Sep 14, 2017 Cologne , Germany Lan Example 10424, token 112
's Eve event in Cologne in 2015. Example 8374, token 48
, Buenos Aires, Cologne , Dublin, Example 17938, token 87
Visit Website ADVERTISEMENT Thanks for watching! Visit Example 4368, token 72
Visit Website ADVERTISEMENT Thanks for watching! Visit Example 2491, token 73
Visit Website ADVERTISEMENT Thanks for watching! Visit Example 11981, token 109
Angel Crespo Bologna 28 £550 Example 955, token 42

(c) Top-activating examples for a feature annotated as "uninterpretable". The specific annotation was `" Whats" > "ADVERTISEMENT Thanks" > "olog" NOT INTERPRETABLE.`

▼ Percentage in band: 0.0024%
events of "Rhinoceros." Example 1525, token 51
<|endoftext|> Neocognitron was Example 4647, token 4
the left, large Neoclassical buildings demanded Example 5234, token 123
stain on "fake Pocahontas'" Example 468, token 62
usting Augusto Pinochet . Example 15770, token 94
horse's mouth. Apocryphal legend has Example 6566, token 86
Dozens of dioceses and Catholic Example 22522, token 64
to south. Such baroclinic zones are Example 8107, token 50
"The Age of Innocence," set in Example 7384, token 103
optical systems. Now modop can be an Example 17167, token 27

▼ Percentage in band: 0.0032%
the lore that Talocan and Jov Example 1892, token 26
The maiden: Innocence; desire; Example 18227, token 76
I'm serious. Rochelle: Boy Example 2061, token 105
Z9 - Smart Choc (@smartch Example 25096, token 82
Made in San Pedro Atocpan" on their Example 1933, token 18
included meetings with Mr. Rockefeller and Example 18303, token 59
. Mechanical instability of monocrystalline and poly Example 15317, token 38
by, TicToc Games has apparently seized Example 10423, token 53
have not detected, unequivocally, a single Example 7147, token 47
the president did not unequivocally say today is Example 18809, token 8

▼ Percentage in band: 0.0024%
able to rent a chocobo to ride. Example 23573, token 105
dictionary are stored in Vocabulary Builder. You Example 21316, token 116
again. We are suffocating.Our peace Example 15063, token 21
supervising attorney of Advocates for Children Services Example 8517, token 56
moth we will be relocating our office to Example 7860, token 47
to participate in the monoculture, let alone Example 2223, token 123
A company relocating to Newark might Example 2659, token 26
help the memory of Slocum come alive and Example 10408, token 59
Main Events, will head Roc Nation's boxing division Example 7812, token 120
plagued by dysfunction. Foc.us "People Example 10445, token 74

(d) Top-activating examples for a feature annotated as "context-free". The specific annotation was `"oc" in middle of words single-token feature.`

Figure 6: Examples of "feature-dashboards" used in the feature interpretation experiments.

layer 9. Specifically, for MLP0 transcoder feature $i$ and MLP10 transcoder feature $j$, this attribution is given by $\left(\mathbf{f}_{\mathbf{dec}}^{(\mathbf{0},\mathbf{i})}\right)^T \left(\mathbf{W}_{\mathbf{OV}}^{(\mathbf{9},\mathbf{1})}\right)^T \mathbf{f}_{\mathbf{enc}}^{(\mathbf{10},\mathbf{j})}$. For each MLP10 transcoder feature, the top ten MLP0 transcoder features were considered. Then, for each MLP0 transcoder feature, the de-embedding score of each YY token for that MLP0 feature was computed. The total de-embedding score of each YY token for an MLP10 feature was computed as the sum of the de-embedding scores of that token over the top ten MLP0 features, with each de-embedding score weighted by the input-invariant attribution of the MLP0 feature. In Figures 5 and 4, the de-embedding scores were scaled and recentered in order to fit on the graph.

The *mean probability difference* metric discussed in the original greater-than work is as follows. Given the logits for each YY token, compute the softmax over these logits in order to obtain a probability distribution over the YY tokens; let $p_y$ denote the probability of the token corresponding to year $y$. Then, the probability difference for a given prompt containing a certain input year $y$ is given by

$\sum_{y'>y} p_{y'} - \sum_{y'\le y} p_{y'}$. The mean probability difference is the mean of the probability differences over all 100 prompts.

# H  Full case studies

## H.1  Classic blind case studies

### H.1.1  Citation feature: `tc8[355]`

First, we checked activations for the first 12,800 prompts in the training data. Using this, we identified the prompt indexed at $(5701, 37)$ as one of 11 prompts for which `tc8[355]` activated above a score of 11.

Path-based analysis on input index $(5701, 37)$ revealed contributions from various tokens, notably `attn7[7]@35` and `attn5[6]@36`. However, we first decided to focus on the current token.

**Current-token features.**  Top de-embeddings for both `tc0[9188]` and `tc0[16632]` were all variants of a semicolon: `;`, `';`, `%;`, and `.;`. We also checked `tc6[11831]@-1` and found that its top contributing features from layer 0 were `tc0[16632]` and `tc0[9188]`: the same two semicolon features. On the basis of this, we concluded that *the final token is a semicolon*.

**Surname features.**  Next we focused on `attn7[7]@35`. Some interpretable features with high attributions through this component included `tc0[13196]@36` (years), `tc0[10109]@31` (open parentheses), `mlp8tc[355]attn7[7]attn0[1]@35` (components of last names), `tc0[12584]@32`: `P`, and `tc0[7659]@34`: `ck`.

Input-independent investigation of `tc6[21046]@35` revealed high contributions from `tc0[16382]` and `tc0[5468]`. feat016382 corresponded to tokens such as `oglu`, `owski`, and `zyk`; `tc0[5468]` corresponded to tokens such as `Burnett`, `Hawkins`, and `MacDonald`. Observing that all of these are (components of) surnames, we decided that *token 35 was likely (part of) a surname*.

**Repeating analysis with prompt** $(6063, 47)$**.**  Top attributions for this prompt once again identified `tc0[9188]`, the semicolon feature from earlier. We filtered our computational paths to exclude this transcoder feature, since we already had a hypothesis about what it was doing. This identified `tc0[10109]@39` and `tc0[21019]@46` as top-contributing features.

The top de-embedding tokens for `tc0[10109]@39` were `(`, `(=`, and `(~`. On the basis of this, we determined that *token 39 was likely an open parenthesis*. Meanwhile, the top de-embedding tokens for `tc0[21019]@46` were `1983`, `1982`, and `1981`. This caused us to conclude that *token 46 was likely a year*.

We noted that, in the previous prompt, the attribution for the year features went through `attn5[6]`, whereas on this prompt it went through `attn2[9]`. We decided to investigate the behavior of `attn5[6]` on this prompt, and found that it was attributing to features `tc0[16542]@11`, `tc0[4205]@11`, and `tc0[19728]@11`. The de-embedding results for these were mixed: `tc0[16542]` were both close-parenthesis features, whereas `tc0[4205]` included citation-related tokens like `Accessed`, `Neuroscience`, and `Springer`.

**Final result.**  We decided that `tc8[355]` was likely a semicolon-in-citations feature and looked at activating prompts. Top-activating prompts included "Res. 15, 241–247; 1978). In their paper, ", "aythamah , 2382; Tahdhīb al-", and "lesions (Poeck, 1969; Rinn, 1984). It". Note that the last of these was prompt $(5701, 37)$, i.e. the first case study we considered.

In general, the top-activating features corroborated our hypothesis, and we did not find any unrelated prompts. We noticed that many of the top activating prompts had a comma before the year in citations, but our circuit analysis never identified a comma feature.

We compared transcoder activations on the prompts "(Leisman, 1976;" and "(Leisman 1976;" and found `tc8[355]` to activate almost identically for both when all preceding MLPs were replaced by transcoders (4.855 and 4.906, respectively) and on the original model (12.484 and 12.13, respectively).

### H.1.2 "Caught" feature: `tc8[235]`.

First, we checked activations for the first 12,800 prompts in the training data. Using this, we identified prompt (8531, 111) as one of 13 prompts for which `tc8[235]` activated above a score of 11.

**Input** (8531, 111). Path analysis revealed that this feature almost exclusively depends on the final token in the input. Input-independent connections to the top-contributing transcoder feature, `tc7[14382]`, revealed the layer-0 transcoder features `tc0[1636]` (de-embeddings: `caught`, `aught`) `tc0[5637]` (de-embeddings: `captured`, `caught`), `tc0[3981]` (`catch`, `catch`) as top contributors.

**Inputs** (6299, 39) **and** (817, 63). For input (6299, 39), we again saw top computational paths depended mostly on the final token. This time, we identified `tc7[14382]` and `tc0[1636]`—both of which were already identified for the previous prompt—as top contributors.

For input (6299, 39) we also observed the same pattern. This caused us to hypothesize that *this feature fires on past-tense synonyms of "to catch."*

**Final result.** Top activating prompts for this feature were all forms of "caught," but the various synonyms, such as "uncovered," were nowhere to be found.

**"Caught" as participle.** Additionally, we noticed that "caught" was used as a participle rather than a finite verb in all top-activating examples. To explore this, we investigated the difference in activations between the prompts "He was caught" and "He caught the ball", and found that the former caused `tc8[235]` to activate strongly (19.97) whereas the latter activated very weakly (0.8145).

When we tested the same prompts while replacing all preceding MLPs with transcoders, we found the difference much less stark: 16.45 for "He was caught" and 9.00 for "He caught the ball". This suggests that transcoders were not accurately modeling this particular nuance of the feature behavior.

Finally, we checked top paths for contributions through the `was` token on the prompt "He was caught" to see whether we could find anything related to this nuance in our circuits. This analysis revealed `attn1[0]@2` as important, and were able to discover mild attributions to transcoder features whose top de-embeddings were `was` and related tokens.

## H.2 Restricted blind case studies

Beyond a simple blind case study, we carried out a number of "restricted blind case studies." In these, all of the rules of a regular blind case study apply, and additionally it is prohibited to look at *input-dependent* information about layer-0 transcoder features.

Since layer 0 features are more commonly single-token features, and in general there is almost no contextual information available for the MLP yet, layer 0 features tend to be substantially more informative about the tokens in the prompt than features in other layers are. Thus, it is often possible to reconstruct large portions of the prompt just from the de-embeddings of which layer 0 transcoder features are active—and, although we never look at these activations directly, they are frequently revealed and analyzed as part of active computational graphs leading to some downstream feature.

By omitting input-dependent information about layer 0 features from our analysis, we must rely more on circuit-level information, and remain substantially more ignorant of the prompts for activating examples. Note that *input-independent* information about layer 0 features can still be used: for instance, we can look at top *input-independent* connections to layer 0 features, and the de-embeddings for those as well—at the expense of not knowing whether those features are active or not.

### H.2.1 Local context feature: `tc8[479]`.

Our first example of a blind case study follows `tc8[479]`, which we fail to correctly annotate through circuit analysis. We include this case study for transparency, and as an instructive example of how things can go awry during blind case studies. First, we measured feature activations over 12,800 prompts and identified 6 prompts that activated above a threshold of 10.

**Input** $(3511, 64)$. For this prompt, path analysis revealed a lot of attention head involvement from many previous tokens. For our first analysis, we chose the path `mlp8tc[479]@-1` `<- attn8[5]@62: 8.1 <- mlp7tc[10719]@62`, since we could look at de-embeddings for `tc7[10719]@62`. Top input-independent connections from `tc7[10719]@62` to layer 0 were `tc0[22324]` and `tc0[2523]`, which had `estimated` and `estimate` as their top de-embeddings, respectively. Thus, we hypothesized that *token 62 is "estimate(d)"*.

Next, we looked at the pullback of `tc8[479]` through `attn8[5]` through `attn7[5]@57`. This revealed top input-independnet connections to `tc0[23855]` (top de-embedding tokens: `spree`, `havoc`, `frenzy`), `tc0[8917]` (took de-embedding tokens: `amounts`, `quantities`, `amount`), and `tc0[327]` (`massive`, `massive`, `huge`). We found this aspect of the analysis to be inconclusive.

The pullback of `tc8[479]` through `attn8[5]` through `attn6[11]@57` revealed connections to `tc0[13184]` (`total`), `tc0[12266]` (`comparable`), and `tc0[12610]` (`averaging`). This led us to believe that *token 57 relates to quantities*.

We found that `tc3[18655]` was a top transcoder feature active on the current token. This showed top input-independent connections to `tc0[11334]` and `tc0[5270]`, both of which de-embedded as `be`. This led us to hypothesize that `tc8[479]` features on phrases like "the amount/total/average is estimated to be...".

**Input** $(668, 122)$. For this prompt, most contributions once again came from previous tokens. The top contributor was `attn8[5]@121`, which had input-independet connections to `tc0[12151]` (`airport`), `tc0[8192]` (`pired`), `tc0[13184]` (`total`), and `tc0[1300]` (`resulted`). This was inconclusive, but this is the second time that `tc0[13184]` has appeared in de-embeddings.

Next, we investigated `attn8[7]@121`: it connected to `tc0[16933]` (`population`), `tc0[14006]` (`kinson`, `rahim`, `LU`, ...), `tc0[19887]` (`blacks`), and `tc0[6821]` (`crowds`). These seemed related to groups of people, but this analysis was also inconclusive.

When we investigated `tc4[18899]@121`, top input-idependent connections to layer-0 features included `tc0[22324]`, which de-embedded to `estimated` again. This was more consistent with the behavior on the previous prompt.

To understand the current-token behavior, we looked at `tc7[13166]@-1`. Top input-indendent connections were `tc0[18204]` (`discrepancy`) and `tc0[14717]` (`velocity`). `tc1[19616]@-1` and `tc3[22544]-1`, both of which also contributed, each had top connections to `tc0[19815]` (`length`). This led us to guess that *this prompt relates to estimated length*.

Next, we looked at previous tokens. One feature, `tc5[10350]@119`, was connected to `tc0[23607]` and `tc0[4252]`, both of which de-embedded to variants of `With`. For the next token, `tc6[15690]@120` was connected to `tc0[22463]` and `tc0[18052]` (both `a`). This updated our hypothesis to something like "with an estimated length."

Further back in the prompt, we saw `tc4[23257]@29` (connected to `tc0[12475]`: `remaining`, `tc0[16996]`: `entirety`).

**Input** $(7589, 89)$. One feature, `tc7[6]@87`, pulled back to `tc0[22324]`, which de-embedded to `estimated`. A following-token feature, `tc1[14473]`, pulled back to `tc0[4746]` (`annual`, `yearly`), and the next-token feature `tc1[12852]@89`, pulled back to `tc0[923]` (`revenue`). Thus, this prompt seemed to end in "estimated yearly revenue."

Estimates for earlier tokens included `tc4[23699]@85` (`tc0[10924]`: `with`), `tc5[6568]@86` (`tc0[1595]`: `a`). This matched the pattern from earlier, where we expected a prompt like "with an estimated length"—but now we expect "with an estimated annual revenue."

Looking at the pulled-back feature `mlp8tc[479]attn3[2]@86`, none of the connections we found to be very informative. This is consistent with patterns observed in other case studies, where pullbacks through attention tended to be harder to interpret.

**Final guess.** On the basis of the above examples, we guessed that this feature fires on prompts like "with a total estimated...". When we viewed top activating examples, we found a number of examples that matched this pattern, especially among the highest total activations. However, for many of the lowest-activation prompts we saw quite different behaviors. Activating prompts revealed that

*this is a local context feature*, which in retrospect may have been apparent through the very high levels of attention head involvement in all circuits we analyzed.

### H.2.2 Single-token `All` feature: `tc8[1447]`

An analysis of the first 12,800 prompts revealed 21 features activating above a threshold of 11. One of these was input $(3067, 79)$. The computational paths for this prompt revealed all contributions came from the final token.

The top attribution was due to `tc7[10932]`, with a top input-independent connection to `tc0[4012]`, which de-embedded to `All`. The next-highest was `tc6[8713]`, which connected to `tc0[6533]`, which de-embedded to `All` (note the leading space). These observations led us to hypothesize *this is probably a simple, single-token feature for "All."*

We also looked at context-based contributions by filtering out current-token features, and found the top attributions to max out at 0.23 (compared to 3.5 from `tc7[10932]@79`). This was quite low, indicating context was probably not very important. Nevertheless, we explored the pullback of `tc8[1447]` through the OV circuit of `attn4[11]@78` and discovered several seemingly-unrelated connections with low attributions. When we pulled back through the OV circuit of `attn1[1]@78` and `attn2[0]@78`, both showed input-independnt connections to features that de-embedded as punctuation tokens. Overall, the context seemed to contribute little, except to suggest that there may be punctuation preceding this instance of `All`.

We repeated this analysis with another input, $(8053, 72)$, and found the same features contributing: `tc7[10932]`, followed by `tc6[8713]`. This led us to conclude *this is a single-token "All" feature.* Top activating examples confirmed this: the feature activated most highly for `All`, then `All`, and finally `all`. Overall, this feature turned out to be quite straightforward, and it was easy to understand its function purely from transcoder circuits.

### H.2.3 Interview feature: `tc8[6569]`

For this feature, we found 15 out of 12,800 prompts to activate above a threshold of 16.

**Input** $(755, 122)$. We started by exploring input $(755, 122)$, which revealed several contributions from other tokens.

We began by looking at components that contributed to the final token. The top feature was `tc7[17738]`, which connected to `tc0[15432]` (variants of `interview`), `tc0[12425]` (variants of `interviewed`), and `tc0[12209]` (tokens like `Transcript`, `Interview`, and `rawdownloadcloneembedreportprint`). The next feature, `tc3[11401]`, was connected to `tc0[15432]` and `tc0[12425]` (same as the previous), as well as `tc0[21414]`, which de-embedded to variants of `spoke`. This raised the possibility that "interview" is being used as a verb in this part of the prompt.

Next, we turned our attention to previous tokens in the context, in hopes that this would clarify the sense in which "interview" was being used. The top attribution for the previous token (121) was through `attn4[11]`. The de-embeddings for top input-independent features were uninformative: `tc0[22216]` seemed to cover variants of `gest`), while `tc0[7791]` covered variants of `sector`. For token 120, pullbacks through `attn2[2]` showed connections to `tc0[10564]` and `tc0[9519]`, both of which de-embedded to variants of `In`. This led us to believe "interview" was in fact being used as a noun, e.g. "in an interview..."

The top attribution for token 119 came through `attn4[9]`, and showed connections to:

- `tc0[625]`: `allegations`, `accusations`, `allegation`, ...,
- `tc0[10661]`: `allegedly`, `purportedly`, `supposedly`, ..., and
- `tc0[22588]`: `reportedly`, `rumored`, `stockp`, ....

The next-highest attribution came through `attn8[5]`, and showed connections to:

- `tc0[4771]`: `Casey`, `Chase`, `depot`, ..., and
- `tc0[5436]`: `didn`, `didn`, `wasn`...

The next-highest was `tc2[5264]@119`, which showed connections to:

- `tc0[5870]`: `unlocks`, `upstairs`, `downstairs`, ...,
- `tc0[14674]`: `said` and variants, and
- `tc0[12915]`: `said` and variants

This led us to believe that *this feature fires on "said in an interview"-type prompts*.

**Input** $(1777, 53)$**.**  Next we tried another prompt, $(1777, 53)$. The top features for the current token were identical to the previous example: `tc7[17738]`, `tc3[11401]`, `tc6[24442]`, and so on.

For the context, we first looked at the pullback of our feature through the OV circuit of `attn2[2]@51`. This showed input-independent connections to `tc0[10564]`, which once again de-embedded to `In`. Next up, `attn4[9]@50`. This feature connected to `tc0[625]`, `tc0[10661]`, and `tc0[22588]`, exactly like before. Recall that these features de-embed to "said" and "allegedly"-type tokens.

Finally, we saw a high attribution from a much earlier token via `attn8[9]@16`. The pullback of our feature through this head showed high input-independent connections to `tc0[14048]`, whose de-embeddings were all variants of `election`.

**Input** $(10179, 90)$**.**  For our last input, we once again found the same transcoder features contributing through the current token. For earlier tokens, we tried:

- `attn2[2]@88`, finding `tc0[10564]` (`In`) again;
- `attn8[9]@86`, finding `tc0[16885]`, which also de-embedded to `elections` despite being a new feature;
- `attn6[20291]@86`, finding `tc0[372]` (`told`); and
- `tc6[20291]@86`, finding `tc0[372]` again.

**Final guess.**  In sum, we decided this feature fires for prompts conveying "told/said in an interview." Top activating examples corroborated this, without any notable deviations from this pattern.

### H.2.4   Four more restricted blind case studies

We present the results of four more restricted blind case studies in Table 3. In the interest of conserving space, only the results of these case studies are presented. However, in the supplemental material attached to this submission, the original Jupyter Notebooks in which the case studies were carried out are provided.

Table 3: The results of four more restricted blind case studies.

| Feature | Final hypothesis | Actual interpretation | Outcome |
|---|---|---|---|
| `tc8[9030]` | Fires on `biology` when in the context of being a subject of study | Fires on scientific subjects of study like `chemistry`, `psychology`, `biology`, `economics` | Failure |
| `tc8[4911]` | Fires on `though` or `although` in the beginning of a clause | Fires on `though` or `although` in the beginning of a clause | **Success** |
| `tc8[6414]` | Largely uninterpretable feature that sometimes fires on Cyrillic text | Largely uninterpretable feature that sometimes fires on Cyrillic text | **Success** |
| `tc8[2725]` | Fires on phrases about not offering things or not providing things. (As a stretch: particularly in legalese context?) | Fires on phrases about not offering things or not providing things, in general | **Mostly a success** |

Table 4: The results of all blind case studies.

| Feature | Type | Final hypothesis | Actual interpretation | Outcome |
|---|---|---|---|---|
| tc8[355] | Blind | Fires on semicolons in the context of academic citations | Fires on semicolons in the context of academic citations | **Success** |
| tc8[1447] | Restricted Blind | Single-token "All" feature | Single-token "All" feature | **Success** |
| tc8[6569] | Restricted Blind | Fires on prompts conveying "told/said in an interview" | Fires on prompts conveying "told/said in an interview" | **Success** |
| tc8[4911] | Restricted Blind | Fires on `though` or `although` in the beginning of a clause | Fires on `though` or `although` in the beginning of a clause | **Success** |
| tc8[6414] | Restricted Blind | Largely uninterpretable feature that sometimes fires on Cyrillic text | Largely uninterpretable feature that sometimes fires on Cyrillic text | **Success** |
| tc8[235] | Blind | Fires on past-tense synonyms of "to catch" | Fires on forms of "caught" | **Mostly a success** |
| tc8[2725] | Restricted Blind | Fires on phrases about not offering things or not providing things. (As a stretch: particularly in legalese context?) | Fires on phrases about not offering things or not providing things, in general | **Mostly a success** |
| tc8[479] | Restricted Blind | Fires on prompts resembling "with a total estimated..." | A local context feature | Failure |
| tc8[9030] | Restricted Blind | Fires on `biology` when in the context of being a subject of study | Fires on scientific subjects of study like `chemistry`, `psychology`, `biology`, `economics` | Failure |

