# OpenReview forum: "Transcoders find interpretable LLM feature circuits"
_NeurIPS.cc/2024/Conference — NeurIPS 2024 poster_

### Official Review · Reviewer_sL5K · 2024-07-10

**Soundness:** 3
**Presentation:** 4
**Contribution:** 3
**Rating:** 6
**Confidence:** 3

**Summary:**

The paper introduces the use of transcoders a tool for mechanistic interpretability as a replacement for SAE. The main difference between SAE and transcoders is that SAE uses autoencoders to take the output of an MLP  and reconstruct it while transcoders (that can also be viewed as an encoder-decoder architecture) take the input of an MLP and reconstruct the output.
The paper compares SAE and transcoders in terms of sparsity (L0 Norm of the features), faithfulness (matching the output of the original MLP), and human interpretability and show that they in par with SAE on multiple-size models.


The paper shows how transcoders can be used to perform weights-based circuit analysis through MLP sublayers by doing the following:
- First identifying attribution between feature pairs:
    - Each feature in a transcoder is associated with two vectors: the i-th row of $W_{enc}$ is the encoder feature vector of feature i, and the i-th column of $W_{dec}$ is the decoder feature vector of feature i. The i-th component of zTC(x) is the activation of feature i.
    - Assuming $l$ is in an earlier layer than $l'$, they  calculate the contribution of feature i in transcoder $l$ to the activation of feature i'
as the product of two terms: the earlier feature’s activation (which depends on the input to the model)  $z^{(l,i)}_{TC}(x^{(l,t)})$ and
 the dot product of the earlier feature’s decoder vector with the later feature’s encoder vector (which is
 independent of the model input).

- Second finding computational subgraphs:
    - On a given input and transcoder feature i', one can determine which earlier-layer transcoder features i are important for causing i' to activate, from the previous step.
    - Once the earlier-layer features i that are relevant to i' are identified, they recurse on i to understand the most important features causing i to activate by repeating this process.
    - Doing so iteratively yields a set of computational paths.
    - These computational paths can then be combined into a computational subgraph, in such a way that each node (transcoder feature), edge, and path is assigned an attribution.


- They defined a de-embeddings:
    - A de-embedding vector for a transcoder feature is a vector that contains the direct effect of the embedding of each token in the model’s vocabulary on the transcoder feature
    - The de-embedding vector for feature i in the layer l transcoder is given by $W_E^Tf_{enc}^{(l,i)}$, where $W_E$ is the model’s token embedding matrix.  This vector gives input-invariant information about how much each possible input token would directly contribute to the feature’s activation.
    - Given a de-embedding vector, looking at which tokens in the model’s vocabulary have the highest de-embedding scores tells us about the feature’s general behavior.


The paper showed a blind case study of how they reverse engineer a feature in GPT2-small and analysed  GPT2-small “greater-than” circuit

**Strengths:**

### Novelty:

- While transducers have been previously introduced this paper is the first to apply them to large models and analyze their performance in various settings.

- The circuit analysis presented in the paper is novel in the sense that it disentangles input-invariant information from input-dependent information.

### Presentation:

- The paper is very well written and easy to follow.

### Application:

- The paper showed how the transducers can be used to reverse engineer features in GPT2-small and they also analyzed GPT2-small “greater-than” circuit comparing transducers to neuronal approach showing that transducers have better performance.

**Weaknesses:**

### Method:

Circuit Analysis: Assuming Transcoders perfectly emulates MLP (which it tries to but as long as faithfulness loss > 0 it doesn't) the circuit analysis graph is between Transcoders in different layers but in the actual model there is multiple-head attention in between each MLP layer which is not accounted for its effect in equation 6.

### Evaluation:

The evaluation was weak at best. The paper did compare with SAE but the only actual quantitative results were the difference between SAE and transcoders in terms of sparsity and faithfulness which only shows the transcoders are on par with SAE. The more important experiment is the interpretability experiment since this is why both methods were developed in the first place for these experiments were not systematic and many of the choices were not justified or explained properly.

**Questions:**

Section 3.2.1:
- For the blind interpretably experiments why was layer 15 in particular chosen?
- From the following sentence: "We recorded  for each feature whether or not there seemed to be an interpretable pattern, and only after examining every feature did we look at which features came from where."
    - How did you identify that there is an "interpretable pattern"?
    - How many people performed this experiment?


Section 4.1:
- For attribution and graph calculation for a given feature i' at layer l' are all others feature at previous layers considered in the graph or only the same feature at the previous layer?

**Limitations:**

Yes, they have.

---

> ### Author Rebuttal · Authors · 2024-08-07
>
> Thank you for taking the time to review our work. We are very glad to see that you recognize the importance of the input-invariant/input-dependent factorization, along with the power demonstrated by transcoders in reverse-engineering feature circuits in an actual model (GPT2-small). We will now address your questions in turn.
>
> > Circuit Analysis: Assuming Transcoders perfectly emulates MLP (which it tries to but as long as faithfulness loss > 0 it doesn't) the circuit analysis graph is between Transcoders in different layers but in the actual model there is multiple-head attention in between each MLP layer which is not accounted for its effect in equation 6.
>
> We show in Appendix D.3 how attribution works through attention heads. We mention this in Line 190 (section 4.1.2), but we agree that this deserves to be moved to the main body of the paper In particular, the contribution of token $s$ at layer $l$ through attention head $h$ to transcoder feature $i’$ at layer $l’ > l$ is given by:
>
> $$ \text{score}^{(l,h)} \left( x_{pre}^{(l,t)}, x_{pre}^{(l,s)} \right) \left( \left( \left( W_{OV}^{(l,h)} \right)^T f_{enc}^{(l’, i’)} \right) \cdot x_{pre}^{(l,s)} \right)$$
>
> Where $\text{score}^{(l,h)}$ denotes the scalar “scoring” function that weights the importance of each source token to the destination token (i.e. the pattern in the QK circuit of attention).
>
> > Evaluation: The evaluation was weak at best. The paper did compare with SAE but the only actual quantitative results were the difference between SAE and transcoders in terms of sparsity and faithfulness which only shows the transcoders are on par with SAE. The more important experiment is the interpretability experiment since this is why both methods were developed in the first place for these experiments were not systematic and many of the choices were not justified or explained properly.
>
> Our primary goal in our work was **to demonstrate that transcoders enable input-invariant circuit analysis that is not possible with current methods using SAEs**. To that end, our sparsity, faithfulness, and interpretability experiments were indeed intended to show that transcoders are on par with SAEs when evaluated on these metrics. Because transcoders also provide additional benefits over SAEs in circuit analysis, this means that **you can obtain all of the benefits for circuit analysis with no penalties compared to SAEs**.
>
> If there are any specific other experiments you would like to see comparing SAEs and transcoders, we are happy to carry them out during the discussion period. But we would like to reiterate that the most important benefit of transcoders is in their circuit analysis capabilities.
>
> > For the blind interpretability experiments why was layer 15 in particular chosen?
>
>
> We provide our rationale in Appendix F:
> “Layer 15 was chosen largely heuristically, because we believed that this layer is late enough in the model to contain complex features, while not so late in the model that features are primarily encapsulating information about which tokens come next.”
>
> > From the following sentence: "We recorded for each feature whether or not there seemed to be an interpretable pattern, and only after examining every feature did we look at which features came from where." How did you identify that there is an "interpretable pattern"? How many people performed this experiment?
>
> Interpretability scores were assigned on the basis of whether it was possible to form a hypothesis about what the feature was doing on the basis of a sample of activating examples. Due to budget constraints that prevented wider-scale experiments from being carried out, only one person performed the interpretable pattern experiment. We would like to emphasize that features from SAEs and transcoders were shuffled together in this experiment, and the subject was unaware of which was which. Please feel free to refer to Figure 6 in the appendix if you’re interested in seeing examples of interpretable, partially interpretable, and non-interpretable features.
>
> > For attribution and graph calculation for a given feature i' at layer l' are all others feature at previous layers considered in the graph or only the same feature at the previous layer?
>
> Equation 6 shows how any upstream feature affects any downstream feature. Because of residual connections in the transformer, this equation is valid for computing the effect of a feature in any previous layer to a feature in any later layer. (In particular, the index $l$ denoting the earlier-layer feature can be any layer less than $l’$ denoting the later-layer feature.)
> It is also worth noting that because transcoders are trained independently on each MLP layer, and because there is not any inter-layer consistency of MLP features anyway, there is no concept of the “same feature” across different layers.
>
> Again, thank you for taking the time to write your detailed review.

---

> > ### Comment · Reviewer_sL5K · 2024-08-12
> > **Thank you for your response**
> >
> > Thank you for your response, my score remains as is.

---

### Official Review · Reviewer_GwPg · 2024-07-10

**Soundness:** 3
**Presentation:** 3
**Contribution:** 4
**Rating:** 7
**Confidence:** 4

**Summary:**

The paper compares transcoders to SAEs for interpreting parts of GPT2-small, Pythia-410M and Pythia 1.4B. Transcoders are SAEs trained to replace a particular MLP in the original model instead of implementing an identity.
They find that the transcoders they train outperform the baseline SAEs they train on the pareto-curve of sparsity, measured by the L0 pseudonorm, with faithfulness, measured by the CE loss of the model recovered.
They describe the attribution technique they use to attribute features to earlier features and other network components.
They apply this attribution technique to interpret feature activations without access to the network inputs. They also use it to analyze the GPT2 "greater than" circuit investigated in https://arxiv.org/abs/2305.00586.

**Strengths:**

I think transcoders are maybe the most important idea for a new interpretability technique to investigate at the moment, and this is the first experimental study of them on language models I’m aware of.  SAEs have a potential issue of finding features that sparsely decompose the input data distribution rather than the computations of the neural network itself. Transcoders seem like the simplest potential solution to this problem since they essentially decompose the operations the network performs instead of its activations.
I like that they showcase using the technique to reverse engineer a circuit in section 4.3.

**Weaknesses:**

In Appendix E line 1050, they state that the baseline SAEs they compare their results against were trained on MLP inputs rather than MLP outputs. I think this is the wrong comparison point. The transcoders only need to reconstruct the new features computed by a single MLP, while an SAE trained on MLP inputs needs to reconstruct all features in the residual stream that are in the preimage of the MLP input matrix. This is potentially a lot more features, which might give the SAEs an inherent disadvantage. Training the SAEs on the MLP output activations instead would be a more appropriate comparison. I think this is the biggest weakness of the paper.


In section 4.3 and figure 5, they use neuron-based analysis as a baseline to compare their transcoder feature-based analysis to. I think this is a very weak baseline. Using e.g. activation PCA components would be more appropriate. This is the main result they have comparing their technique to standard techniques outside the cluster of techniques based on SAEs/sparse decoding, so I think not using an appropriate baseline here is not great. I don't think it is a very major flaw, because this is mainly a paper that takes it as given that SAEs are interesting and attempt to improve on them, rather than a paper trying to justify the sparse decoding approach to interpretability.


They describe the attribution method between transcoder features they use as a 'new technique', which I don’t think is justified. It's effectively the attribution patching technique https://arxiv.org/abs/2310.10348 , https://arxiv.org/abs/2403.00745, used e.g. in https://arxiv.org/abs/2403.19647. Transcoders just have the advantage of making it work more elegantly. Any ReLU MLP network attribution patching is applied to would yield equation 6 as well.


In section 3.2.1 and table 1, manual comparison on fifty features isn’t a very large sample size for this sort of analysis. Supplementing this with e.g. a larger set of LLM generated autointerpretability scores would have been better, as was done in https://arxiv.org/abs/2309.08600. Personally, I don’t think not investing more effort into this is a major flaw, because I think reconstruction error and L0 are more important metrics than human interpretability scores.


In Figure 2, they present reconstruction scores for transcoders and SAEs in terms of raw CE-loss recovered scores. I think it would have been good for the presentation to also show the recovered performance in terms of the compute required to train a model with the same CE score, as suggested in https://arxiv.org/abs/2406.04093. Raw CE scores can make the performance gap to the model look smaller than it is, since the difficulty of further reducing CE loss grows as models become more performant.

**Questions:**

Why train the baseline SAEs on the MLP inputs instead of the outputs? This way, the SAEs need to reconstruct all features in the residual stream within the MLP preimage, while the transcoder only needs to reconstruct the new features computed by the MLP.
In section 4.3, why use neuron-based analysis as a baseline? Neurons tend to be very brittle under ablations in my experience. Why not e.g. activation PCA components?

**Limitations:**

All addressed.

---

> ### Author Rebuttal · Authors · 2024-08-07
>
> Thank you for taking the time to write your thoughtful review and for recognizing the importance of transcoders to the broader mechanistic interpretability research program. We are also very excited to continue working on transcoders and seeing how far they can go.
>
> Now, we would like to respond to your various questions.
>
> > In Appendix E line 1050, they state that the baseline SAEs they compare their results against were trained on MLP inputs rather than MLP outputs. I think this is the wrong comparison point. [...] Training the SAEs on the MLP output activations instead would be a more appropriate comparison. I think this is the biggest weakness of the paper.
>
> This is a piece of feedback that we received from other readers as well post-submission, and we have responded by repeating these experiments for SAEs trained on the MLP output activations. The results were similar: transcoders performed on-par with or better than the SAEs by the sparsity/fidelity metrics. Our updated figure is attached to our author rebuttal in our PDF.
>
> > In section 4.3 and figure 5, they use neuron-based analysis as a baseline to compare their transcoder feature-based analysis to. I think this is a very weak baseline. Using e.g. activation PCA components would be more appropriate. This is the main result they have comparing their technique to standard techniques outside the cluster of techniques based on SAEs/sparse decoding, so I think not using an appropriate baseline here is not great. I don't think it is a very major flaw, because this is mainly a paper that takes it as given that SAEs are interesting and attempt to improve on them, rather than a paper trying to justify the sparse decoding approach to interpretability.
>
> Thank you for the suggestion. We avoided looking at PCA components on the activations because to some extent, they suffer from the same problem as SAEs when applied to this task: namely, they do not address the computation carried out by the MLP, but only deal with the intermediate activations of those computations. In carrying out our investigation of the “greater-than” circuit, we wanted to understand how MLP10 in GPT2-small maps year tokens represented in the MLP input space to MLP outputs that boost certain logits. Because PCA (just like SAEs) only handles the intermediate activations of this computation, it is unable to explain this mapping in an input-invariant way. (Note that the original “greater-than” circuit paper also does not use PCA, presumably for similar reasons.)
>
> > They describe the attribution method between transcoder features they use as a 'new technique', which I don’t think is justified. It's effectively the attribution patching technique [...] Transcoders just have the advantage of making it work more elegantly. Any ReLU MLP network attribution patching is applied to would yield equation 6 as well.
>
> We are aware of this, and had already modified the paper post-submission to include a footnote explaining that this is a special case of the classic “input-times-gradient” method for computing attributions. (As you note, this ends up being equivalent to attribution patching when looking at a single transcoder feature, rather than a vector of feature activations.)
>
> What we consider to be a contribution of ours is our recognition that input-times-gradient, when applied to ReLU MLPs such as transcoders, does factorize into an input-dependent and an input-invariant term – i.e. the gradient is constant. Furthermore, something that we like to underscore when thinking about Equation 6 is that for transcoders (and this is not true in general for ReLU networks), the input-dependent term (the feature activation) is interpretable (because feature activations are largely interpretable).
>
> Based on your comment, we can modify the paper to further highlight the relationship between Equation 6 and existing attribution methods.
>
> > In section 3.2.1 and table 1, manual comparison on fifty features isn’t a very large sample size for this sort of analysis. Supplementing this with e.g. a larger set of LLM generated autointerpretability scores would have been better, as was done in https://arxiv.org/abs/2309.08600. Personally, I don’t think not investing more effort into this is a major flaw, because I think reconstruction error and L0 are more important metrics than human interpretability scores.
>
> Thank you for the suggestion. Indeed, we did recognize that fifty features each from a transcoder and SAE was a small sample size, but unfortunately, resource limitations prevented us from carrying out LLM-generated feature scoring on far more features. That said, note that our initial interpretability comparison was primarily intended to provide initial evidence for transcoders being approximately as interpretable as SAEs, and as such, able to be used in furthering our main goal of performing circuit analysis with them.
>
> > In Figure 2, they present reconstruction scores for transcoders and SAEs in terms of raw CE-loss recovered scores. I think it would have been good for the presentation to also show the recovered performance in terms of the compute required to train a model with the same CE score, as suggested in https://arxiv.org/abs/2406.04093. Raw CE scores can make the performance gap to the model look smaller than it is, since the difficulty of further reducing CE loss grows as models become more performant.
>
> Thank you for this suggestion – this is a very interesting idea that we had not previously considered. We can update the figures to include an additional set of y-axis ticks reflecting the amount of compute that a Chinchilla-optimal model would require to achieve the same loss.
>
> > Questions: …
>
> We believe that all of the questions have been addressed above; please let us know if you have any more.
>
> Again, thank you so much for your review.

---

> > ### Comment · Reviewer_GwPg · 2024-08-13
> > **Response**
> >
> > Thank you to the authors for their responses and updates.
> >
> > The new plots comparing MLP-out SAEs fully address what I considered the primary weakness of the paper. All the other weaknesses I noted are less important. This raises my rating of the paper's soundness.
> >
> > > "What we consider to be a contribution of ours is our recognition that input-times-gradient, when applied to ReLU MLPs such as transcoders, does factorize into an input-dependent and an input-invariant term – i.e. the gradient is constant. Furthermore, something that we like to underscore when thinking about Equation 6 is that for transcoders (and this is not true in general for ReLU networks), the input-dependent term (the feature activation) is interpretable (because feature activations are largely interpretable)."
> >
> > The gradient is only constant on the subset of data where the target feature is active. Outside of that subset, it is 0. How and when the threshold is crossed and the target feature becomes active depends on the interactions between the input features. This information is important for understanding the network and cannot be straightforwardly read off from what you call the input-independent term.
> >
> > Meaning the input-dependent term is actually the input ‘feature’ activation times either 1 if the target feature is on, or 0 if the target feature is off. This quantity may be a lot less interpretable than a feature activation.

---

> > > ### Author Response · Authors · 2024-08-14
> > >
> > > Thank you for calling attention to this subtlety. As you note, it is true that the input-times-gradient of the post-ReLU target feature activation with respect to an input feature activation does include an extra input-dependent term that is 1 when the target feature fires and 0 when it does not. But when considering feature attributions through a single transcoder, if we use the pre-ReLU target feature activation instead of the post-ReLU target feature activation, then this extra term does go away. One reason to use the pre-ReLU target activation is that even when the target feature isn't active, you can still use the attributions from Equation 6 to understand which input features are contributing the most to causing the target feature to be *inactive* (i.e. which input features have the most negative attributions). You can also look at the input-invariant scores for each input feature to understand which input features would be most effective in "flipping" the target feature and causing it to activate. Because this "pre-ReLU input-times-gradient" approach allows us to reason in this manner about even inactive target features, we personally prefer it.
> > >
> > > We do agree with your more general point that the interactions between input features can be complex, especially when considering feature attributions through multiple transcoders (i.e. computational paths of path length at least 2), as in this setting, the additional binary input-dependent factor is necessary. We believe that finding interpretable ways to characterize these interactions will be a fruitful area of future research. But for now, we are glad that transcoders have provided the initial groundwork for such questions to be asked.

---

### Official Review · Reviewer_xRyp · 2024-07-13

**Soundness:** 3
**Presentation:** 2
**Contribution:** 3
**Rating:** 7
**Confidence:** 4

**Summary:**

Sparse Autoencoders (SAEs) have been used for interpretability of transformer neural networks. SAEs take the output activations of MLPs within transformers and learn are trained to re-construct those outputs: $SAE(f(x)) \approx f(x)$. In this submission, the authors study using transcoders for a similar task. Unlike SAEs, transcoders are trained to imitate the MLPs directly: $TC(x) \approx f(x)$. The authors demonstrate via experiments and blind trials that the performance of transcoders is competitive with and sometimes exceeds that of SAEs.

**Strengths:**

- Transcoders are a natural and unexplored extension of the SAE architecture that perform very comparably to traditional SAEs.
- Deriving the decomposition into input dependent / invariant features is novel and addresses a significant challenge of interpretability. On its own this warrants future exploration.
- The feature case studies are a valuable contribution to the literature.
- Testing interpretability via blind feature tests is innovative.

**Weaknesses:**

- The concrete comparisons to traditional SAEs are not sufficiently thorough. Experiments demonstrate that the KL divergence of transcoders are competitive compared to SAEs, but my understanding of the SAE literature is that it is still an open question as to how to judge the quality of an SAE. There are, however, other questions that are more concrete but are unaddressed. e.g. how easy/stable are transcoders to train compared to SAEs? Do transcoders suffer issues with "dead neurons" when the width is scaled up? Do transcoders show similar "feature splitting" effects when the width is scaled up? A more holistic comparison to SAEs would strengthen the submission significantly.
- While performing a blind test of the interpretability of different features is novel, I worry that it is prematurely rigorous.
- One of the implicit assumptions of the blind interpretability experiment is that the features learned by transcoders and SAEs are different. Exactly what a "feature" is is still being worked out in the literature, but if transcoders and SAEs truly do learn distinct sets of "features" that is an important fact that should not have been left out of the paper. If the authors did not investigate the extent of the overlap between the features that SAEs and transcoders learn that is unfortunate, but seems like a promising avenue for future investigations.

**Questions:**

- What is the effect of different levels of $L_1$ penalty on the features that the transcoder learns? Do transcoders also experience "feature splitting" as the width of the transcoder goes up?
	- Are transcoders easy to train? Are the learning dynamics stable? Do they experience issues with "dead neurons"?
	- Does the input invariant / dependent construction not work for the features that SAEs learn?
	- When looking at
	- Do SAEs and transcoders trained on the same layer and same data learn the same or similar features?
	- Have any attempts been made to perform a similar analysis for other circuits? e.g. the [IOI](https://openreview.net/forum?id=NpsVSN6o4ul) or [Gendered Pronoun circuits](https://cmathw.itch.io/identifying-a-preliminary-circuit-for-predicting-gendered-pronouns-in-gpt-2-smal)?

**Limitations:**

The authors adequately address the submission's limitations.

---

> ### Author Rebuttal · Authors · 2024-08-07
>
> **TL;DR**: Thank you so much for taking the time to review our work. We were glad to see that you enjoyed our blind case studies and that you recognized the importance of the input-invariant/input-dependent decomposition of attributions. We performed some experiments in response to your questions about learned features, but we’d like to emphasize that our primary goal in this work was to use transcoders to perform input-invariant circuit analysis that SAEs can’t do.
>
> We’ll now address your concerns and questions in order.
>
> >The concrete comparisons to traditional SAEs are not sufficiently thorough [...] There are, however, other questions that are more concrete but are unaddressed. e.g. how easy/stable are transcoders to train compared to SAEs?
>
> We did not notice any meaningful differences in the difficulty of training transcoders compared to SAEs. (Note that we primarily follow the same training procedure as used for training SAEs, as our pipeline is adapted from SAE training pipelines.)
> The reason why we do not address this in the paper is because we did not find the experience of training transcoders to be noteworthy compared to training SAEs, and because training transcoders was instrumental to our primary goal of utilizing transcoders for circuit analysis.
>
> >Do transcoders suffer issues with "dead neurons" when the width is scaled up? Do transcoders show similar "feature splitting" effects when the width is scaled up?
>
> In response to your question, we quickly analyzed the number of dead neurons in transcoders versus MLP-out SAEs with similar average sparsities (L0s), and found no clear winner when it comes to whether transcoders or SAEs have more dead neurons. (See the author rebuttal PDF for a graph.) As stated earlier, this investigation of training dynamics was not a core focus of ours. In any event, any dead neurons in the transcoders that we trained did not seem to affect fidelity or interpretability.
>
> As for feature splitting, it seems that there are currently no rigorous metrics agreed upon by the mechanistic interpretability community for quantifying feature splitting. That said, we did train a pair of GPT2-small transcoders with expansion factors 32 and 64, and found that ~27% of live smaller-transcoder features were very similar (cossim > 0.95) to at least one feature in the larger transcoder. This seems to indicate some amount of feature splitting, but we find it difficult to interpret this number further.
>
> >While performing a blind test of the interpretability of different features is novel, I worry that it is prematurely rigorous.
> >One of the implicit assumptions of the blind interpretability experiment is that the features learned by transcoders and SAEs are different [...] If the authors did not investigate the extent of the overlap between the features that SAEs and transcoders learn that is unfortunate.
>
> Please note that our intent with these blind tests was to show that transcoder features are “up to par” with SAE features in interpretability, regardless of whether or not the features are different. This is because our main goal with transcoders is to use them to perform input-independent circuit analysis that can’t be carried out with SAEs, so we simply wanted to make sure that in achieving this, no penalties to interpretability were accrued. Importantly, this means that we weren’t intending to make any claims about whether transcoders learn “better” features than SAEs; rather, we intended to show that whatever features the transcoders do learn are equally interpretable to those learned by SAEs, without making any assumptions on the types of features being learned by SAEs versus transcoders.
>
> > What is the effect of different levels of L1 penalty on the features that the transcoder learns?
>
> We did not investigate differences in features learned by transcoders at different sparsity levels, in part because we are primarily interested in the circuits learned by transcoders – and because of a lack of accepted methods for rigorously characterizing transcoder/SAE features learned at different sparsity levels. To our knowledge, this is a current lacuna in the mechanistic interpretability community’s collective understanding.
>
> > Does the input invariant / dependent construction not work for the features that SAEs learn?
>
> **This construction does not in fact work for SAEs; we believe that this is the primary advantage of transcoders.** This is not because of the features that SAEs learn per se, but because SAEs fail to bypass the MLP sublayer whose nonlinearity prevents such a construction from being applied. In contrast, when computing attributions, transcoders explicitly bypass the MLP sublayer.
>
> More formally, the input-invariant/input-dependent factorization for transcoder attributions is a special case of the classic “input-times-gradient” attribution method. For transcoders, the “input” term is input-dependent, but the “gradient” term is constant. But for SAE attributions, both the “input” term and the “gradient” term are input-dependent. (We would like to go into more mathematical detail here, but the rebuttal character limit prevents us from this.)
>
> > When looking at [sic]
>
> It seems that this comment may have been cut off. If you can recall what you intended to write, we will be happy to reply during the discussion period.
>
> > Do SAEs and transcoders trained on the same layer and same data learn the same or similar features?
>
> We performed some brief exploratory experiments a while back investigating this question on one-layer models. However, we have not continued to pursue this, as it was tangential to our primary goal of using transcoders for circuit analysis, although we do find it very interesting.
>
> > Have any attempts been made to perform a similar analysis for other circuits? e.g. the IOI or Gendered Pronoun circuits?
>
> We have not carried these out yet, but we agree this is a promising direction for future research.

---

> > ### Comment · Reviewer_xRyp · 2024-08-13
> >
> > Thank you for your thoughtful responses. Upon consideration I have increased my score.

---

> ### Author Response · Authors · 2024-08-12
>
> Before this phase of the discussion period ends, we wanted to check in with the reviewer on whether we have addressed your concerns with our work?

---

### Official Review · Reviewer_Q9a1 · 2024-07-15

**Soundness:** 3
**Presentation:** 3
**Contribution:** 2
**Rating:** 6
**Confidence:** 4

**Summary:**

Motivated by prior claims on how MLP sublayers make interpretability challenging (arguably due to their extremely dense nature), this paper proposes "Transcoders", a protocol that is in spirit similar to Sparse Autoencoders (SAEs). Specifically, a Transcoder aims at "faithfully" matching the output of a layer (in this case, an MLP layer); in contrast, an SAE aims to reconstruct the input to a layer. Results show Transcoders are similarly effective when compared to SAEs (in terms of model loss). Authors follow a blind evaluation pipeline to see if Transcoders yield interpretable features, whereby hypotheses are made for what a feature means and then posthoc the hypothesis is tested by looking at input samples.

**Strengths:**

I enjoyed reading the paper. It's written fairly clearly and the targeted problem is well described, though some more expansion on how MLPs are a challenge for interpretability would help (see weaknesses). The blind evaluation protocol was really good to see---I appreciated authors' efforts in stress testing the limits of their approach (this is how good science should go :)).

**Weaknesses:**

While I like the paper, I think it needs a bit more work to be ready for acceptance. To that end, following are a few suggestions.

- **Expand on challenges in MLP interpretability.** This is the bulk of the motivation of the paper and, from what I can gather, is grounded in off-hand references in two prior works. I don't think those works thoroughly describe what the challenges in fine-grained circuit analysis of MLP sublayers are. It is fine to use those works as motivation, but I think the paper should arguably start off with an experiment to demonstrate the challenge more thoroughly, given that prior work hasn't done that. In fact, I would argue that your own results with SAEs show similar interpretability may be achievable as found via Transcoders (since SAEs perform similar in terms of model loss). From that perspective, it isn't established if the question being addressed is sufficiently a challenge

- **Analysis is correlational, lacking a quantitative confirmation and causal interventions.** While I really like the blind evaluation pipeline, there are two missing elements in the results per my opinion. These render the current analysis somewhat informal. *First*, once a hypothesis is formed, one should ideally run an algorithmic test to see on what proportion of a dataset where the hypothesis predicts a feature will activate does the feature activate. From what I gathered, the results merely involve "eyeballing" a few inputs to see whether the hypothesis holds. *Second*, the experiments are correlational, not causal. Ideally, one would systematically intervene on the input and see if the claimed hypothesis breaks in a predictable way. For example, when one identifies a feature activates for semicolons being present in the input, then if that input is kept the same but the semicolon is dropped or another relevant punctuation is added, does the feature not activate or at least activate less?

- **Figure 2.** I found Figure 2 to be rather unintuitive. The axes labels are quite small, so I thought the loss increases along y-axis and hence Transcoders underperform SAEs, but then realized that the axis is decreasing along y-axis. I would have preferred a more standard plotting schematic, or at least larger labels to help avoid misinterpretation.


**Post rebuttals update.** Taking into account the context provided by the authors and reading through the paper again, I am happy to raise my score. My primary concern was the paper was pitched to solve a very specific problem, i.e., interpreting MLPs. Authors' response indicates this was merely part of the problem they set out to address: identifying circuits in an input-invariant manner, since the interaction between circuit discovery and input-sensitive features makes that pipeline difficult. Overall, I understand the pitch much better now. The paper intro (and other relevant parts) should be updated to accurately reflect authors' motivation. I do not see these rewrites as a major challenge, so I'd be happy if the paper is accepted.

**Questions:**

- From what I can gather, the "de-embedding" tool is equivalent to logit lens? Can you expand on what's different? I understand this is not a core contribution; just wondering why a different term was used.

- I am still struggling to understand the benefit of Transcoders over SAEs. Can you expand on this more? It's good to define alternative tools, but unsure if this was a sufficiently different tool and led to much benefits.

**Limitations:**

See weaknesses.

---

> ### Author Rebuttal · Authors · 2024-08-07
>
> **TL;DR**: Thank you for taking the time to write this detailed review. We are glad to see that you enjoyed the presentation of our work, along with our interpretability study and blind case studies. Our initial paper was insufficiently clear that the main goal of transcoders is to enable input-invariant circuit analysis, which is impossible with any existing SAE-based feature circuit methods. We believe that most of your concerns stem from this, and as such, if we have addressed them, we hope that you might be willing to raise your score.
>
> We’ll now respond inline to specific concerns:
>
> >Expand on challenges in MLP interpretability. This is the bulk of the motivation of the paper and, from what I can gather, is grounded in off-hand references in two prior works. I don't think those works thoroughly describe what the challenges in fine-grained circuit analysis of MLP sublayers are. [...]
>
> Thank you for pointing this out---we realize now that we might not have properly explained the motivation behind the problem. In essence, **the primary problem in fine-grained circuit analysis through MLPs is a lack of input-invariant methods for distinguishing between local and global behavior**. The two prior works that we cited represent different attempts at performing this fine-grained circuit analysis, both of which fail to address this problem. Marks et al. 2024 use causal methods (such as attribution patching), which only yield the importance of features on a single input. Dunefsky et al. 2024 attempt to approximate MLPs with their gradients, but their method has the same shortcoming, because MLP gradients are very different on different inputs. To our knowledge, no current works on performing circuit analysis involving MLPs with SAEs avoid this problem, and we fear that this is an issue inherent to all such methods.
>
> To make all of this clearer to future readers, we plan to add to the paper an explicit, mathematically-motivated explanation of how these SAE-based methods fail.
>
> >In fact, I would argue that your own results with SAEs show similar interpretability may be achievable as found via Transcoders (since SAEs perform similar in terms of model loss). From that perspective, it isn't established if the question being addressed is sufficiently a challenge.
>
> We hope that it is now clearer, in the context of our answer to the previous question, why this is not the case. In particular, since our goal is to use transcoders to perform input-invariant circuit analysis, we view it as a positive that transcoders and SAEs have similar interpretability/model loss – *because this means that we can substitute transcoders for SAEs and obtain all of the benefits of input-invariant circuit analysis without incurring any penalties*.
>
> >Analysis is correlational, lacking a quantitative confirmation and causal interventions. While I really like the blind evaluation pipeline, there are two missing elements in the results per my opinion [...] First, once a hypothesis is formed, one should ideally run an algorithmic test to see on what proportion of a dataset where the hypothesis predicts a feature will activate does the feature activate.
>
> The main reason why we did not perform any algorithmic tests of our feature hypotheses is that the features that we were investigating were mostly too complex to admit any such proxies. Whereas, for example, the original Towards Monosemanticity paper was able to cherry-pick features (such as a base64 feature or Arabic feature) that could easily be captured in such a manner, our blind case study pipeline meant that we did not know if a feature would be easily “proxied” until after having carried out the entire case study.
>
> Indeed, while proxies can be useful for quantitatively evaluating the interpretability of an individual feature, it is worth noting that our goal with blind case studies was broader than this: we view blind case studies as an example of a nontrivial circuit-level interpretability task. To that end, we think that blind case studies act as a useful evaluation of how well a circuit analysis method would fare in a more real-world setting.
>
> >Second, the experiments are correlational, not causal. Ideally, one would systematically intervene on the input and see if the claimed hypothesis breaks in a predictable way. For example, when one identifies a feature activates for semicolons being present in the input, then if that input is kept the same but the semicolon is dropped or another relevant punctuation is added, does the feature not activate or at least activate less?
>
> Although space constraints prevented us from highlighting this in the main body, we have performed such tests of our hypotheses which can be found in the appendices. Please refer to the end of Appendix H.1.1 and Appendix H.1.2 for examples of this in two of our case studies.
>
> >Figure 2. I found Figure 2 to be rather unintuitive [...]
>
> Thank you for pointing this out. In general, the orientation of our axes was inspired by papers such as the DeepMind Gated SAEs paper. However, we didn’t realize that our axis label sizes would cause a problem, and as such, we will make them bigger.
>
> > From what I can gather, the "de-embedding" tool is equivalent to logit lens? Can you expand on what's different?
>
> The de-embedding is a “reverse” logit lens: in the de-embedding, we multiply the encoder vector of the feature by the transpose of the model embedding matrix, in order to understand which tokens in the input vocabulary space cause the feature to activate.
>
> >I am still struggling to understand the benefit of Transcoders over SAEs. Can you expand on this more? It's good to define alternative tools, but unsure if this was a sufficiently different tool and led to much benefits.
>
> We hope that the preceding discussion has clarified our position on this, which is that we think **the main benefit of transcoders is the input-invariant circuit analysis they enable**.

---

> ### Comment · Reviewer_Q9a1 · 2024-08-12
>
> Thank you to the authors for their response.
>
> I have a quick follow up question. Authors say in the paper that their goal was to design a tool for MLPs' interpretability, with statements that remark upon difficulties in analyzing them. In the rebuttals response though (both for my comments and in the global response), authors say their goal in this paper was to perform an input-invariant circuit analysis. I am struggling to reconcile these two points. It would help if the authors can expand on this bit. To be more precise, I would like to know:
>
> 1. Why is MLPs' interpretability deemed difficult? Why can we not use SAEs for this purpose, and why do we need a new tool to this end?
>
> 2. If I do analyze MLPs' features via SAEs, what blockades will I face such that input-invariance becomes a desirable property?
>
> 3. If the argument is input-invariance is a generally useful property (which I would buy), then why should we not use Transcoders for representations extracted from any unit in the model? Why is the authors' pitch that Transcoders are specifically motivated to address some challenge (unclear which challenge) faced in MLPs' interpretability?

---

> > ### Author Response · Authors · 2024-08-12
> >
> > Thank you for engaging with our rebuttal. We are happy to further clarify our view on the role of MLPs in circuit analysis.
> >
> > > Why is MLPs' interpretability deemed difficult? Why can we not use SAEs for this purpose, and why do we need a new tool to this end?
> >
> > Unlike transcoders, **SAEs cannot tell us about the general input-output behavior of MLP layers.** In particular, doing this with SAEs would entail computing the attribution of pre-MLP SAE features to post-MLP SAE features: how much the activation of the post-MLP feature depends on the pre-MLP feature **when mapped through the MLP.** Standard methods for computing attributions are causal patching (which inherently only gives you information about local MLP behavior on a single input) and methods like input-times-gradient or attribution patching (which are equivalent in this setting). To see why these methods are unable to yield information about the MLP’s general behavior, let’s try to use input-times-gradient to compute the attribution of an earlier-layer feature to a later-layer feature.
> >
> > Let $\mathbf{z}$ be the activation of an earlier-layer feature and $\mathbf{z’}$ be the activation of the later-layer feature; similarly, let $\mathbf{y}$ be the activations of the MLP at layer $l’$. Then the input-times-gradient is given by:
> > $$
> > \mathbf{z} (\frac{\partial \mathbf{z’}}{\partial \mathbf{z}}) =
> > \mathbf{z} (\frac{\partial \mathbf{z’}}{\partial \mathbf{y}} \frac{\partial \mathbf{y}}{\partial \mathbf{z}}).$$
> >
> > Unfortunately, not only is the $z$ term input-dependent, but the $\frac{\partial \mathbf{z’}}{\partial \mathbf{z}}$ term **is input-dependent as well!** This is because the $\frac{\partial \mathbf{y}}{\partial \mathbf{z}}$ term—that is, the gradient of MLP activations with respect to the feature activation at the MLP input—is input-dependent. (And this is to be expected, since MLPs are highly nonlinear: of course their gradients would change with respect to their input.)
> >
> > This means that **we cannot use SAEs to understand the general behavior of MLPs on various inputs.** In contrast, with transcoders, we can use the input-invariant term to understand the behavior of the MLP on all inputs.
> >
> > > If I do analyze MLPs' features via SAEs, what blockades will I face such that input-invariance becomes a desirable property?
> >
> > To address why input-invariance is desirable, consider the following example: say that you have a post-MLP SAE feature and you want to see how it is computed from pre-MLP SAE features. Doing e.g. patching on one input shows that a pre-MLP feature for Polish last names is important for causing the post-MLP feature to activate. But on other inputs, would features other than the Polish last name feature also cause the post-MLP feature to fire (e.g. an English last names feature)? Could there be other inputs where the Polish last names feature fires but the post-MLP feature doesn’t? We can see that without input-invariance, it is difficult to make general claims about model behavior.
> >
> > > If the argument is input-invariance is a generally useful property (which I would buy), then why should we not use Transcoders for representations extracted from any unit in the model? Why is the authors' pitch that Transcoders are specifically motivated to address some challenge (unclear which challenge) faced in MLPs' interpretability?
> >
> > In fact, we can use solely transcoders for circuit analysis; this is what we do in our case studies. (This involves taking pullbacks of transcoder features by attention OV matrices, as discussed in the appendix.) In practice, though, using SAEs trained on different activation points (in conjunction with MLP transcoders) might yield features that are more interpretable for those activation points. But importantly, note that the challenge transcoders intend to solve is not MLP interpretability per se; it is input invariant circuit analysis through MLPs.
> >
> > We hope that this has helped to clarify, and if you have any other questions, we would be happy to answer.

---

> ### Author Response · Authors · 2024-08-13
> **Elaboration on the difficulty of input-invariant circuit analysis of MLPs**
>
> Some further thoughts on your first question that we hope may be helpful:
> In circuit analysis work, a key goal is to decompose layers, large and complex objects, into independent components, so we can find a sparse set of components that matter for a given task. For attention layers, the layer naturally breaks down into independently varying attention heads. In prior work finding circuits, researchers often analyse individual heads, eg induction heads [1] or name mover heads [2]. Crucially, this works because attention heads often seem to be interpretable, at least in the context of a given task.
>
> In MLP layers, however, things are harder. Though MLP layers decompose into independently varying neurons, these neurons are much harder to work with for circuit analysis than heads, as neurons are often polysemantic [3], i.e. activate for many seemingly unrelated things, and often many neurons seem relevant for a given task [4].  Prior work on circuit analysis that looks at MLPs [5, 6] has largely failed to find more granular decompositions just studying an entire MLP layer (though some works have made some progress [7]). Thus, by default, circuit analysis needs to either include or exclude an entire MLP layer, rather than being able to zoom in further. MLPs represent at least 60% of the parameters in models like GPT-2, so this lack of fine-grained analysis is a major roadblock to circuit analysis.
>
> SAEs help by decomposing the *output* of the MLP layer, which can help causal intervention based circuit analysis [8], but SAE features are often dense combinations of many neurons [9], so we must still consider many neurons in the MLP layer, whose behaviour will vary depending on the input, preventing input-invariant circuit analysis.
>
> Transcoders solve this problem by *replacing* the MLP layer with a sparser and more interpretable replacement layer. Transcoders features are easier to work with for circuit analysis, because they break the MLP layer down into fine-grained computations that do *not* depend on MLP neurons, which are often interpretable and can be studied independently as part of a circuit analysis. Transcoder features are computed directly from the MLP input (projecting the MLP input onto the encoder vector followed by a bias and a ReLU), allowing us to decompose the MLP layer better for circuit analysis in an input-invariant way.
>
> Please let us know if there is anything further that we can clarify. If we have successfully addressed your concerns, we hope that you may consider raising your score.
>
> [1] In-context Learning and Induction Heads. Olsson et al. https://arxiv.org/abs/2209.11895
>
> [2] Interpretability in the Wild: a Circuit for Indirect Object Identification in GPT-2 small. Wang et al. https://arxiv.org/abs/2211.00593
>
> [3] Softmax Linear Units. Elhage et al. https://transformer-circuits.pub/2022/solu/index.html
>
> [4] Finding Neurons in a Haystack: Case Studies with Sparse Probing. Gurnee et al. https://arxiv.org/abs/2305.01610
>
> [5] Does Circuit Analysis Interpretability Scale? Evidence from Multiple Choice Capabilities in Chinchilla. Lieberum et al. https://arxiv.org/abs/2307.09458
>
> [6] Fact Finding: Attempting to Reverse-Engineer Factual Recall on the Neuron Level. Nanda et al. https://www.alignmentforum.org/posts/iGuwZTHWb6DFY3sKB/fact-finding-attempting-to-reverse-engineer-factual-recall
>
> [7] How does GPT-2 compute greater-than? Hanna et al. https://arxiv.org/abs/2305.00586
>
> [8] Sparse Feature Circuits: Discovering and Editing Interpretable Causal Graphs in Language Models. Marks et al. https://arxiv.org/abs/2403.19647
>
> [9] Open Source Replication & Commentary on Anthropic's Dictionary Learning Paper. Neel Nanda. https://www.alignmentforum.org/posts/fKuugaxt2XLTkASkk/open-source-replication-and-commentary-on-anthropic-s

---

> > ### Comment · Reviewer_Q9a1 · 2024-08-13
> >
> > Thank you for the response! The answers provide sufficient context for me to understand authors' motivation. I do still have apprehensions whether the motivation introduced in the current version of the paper and what the authors have clarified in rebuttals completely align. I'll reread the paper with the additional rebuttals context this week and try to assess if a camera-ready revision will suffice to make any changes that might be necessary for clarity of motivation, or whether a new submission should be made. I'll update my score if the former seems feasible.

---

> > > ### Author Response · Authors · 2024-08-13
> > > **Proposed edits overview**
> > >
> > > We are very glad to hear that we have been able to better convey the problem that we are trying to address, and we are grateful that you are willing to consider raising your score.  For your information, the following is a set of edits that we plan on making to the paper in order to clarify all of this, based on both your feedback and that of other readers:
> > > * Edit intro after line 37 to include “This means that SAEs cannot tell us **about the general input-output behavior of an MLP across all inputs.**”
> > > * Then, after line 37, include a brief discussion of why input-invariance is important – refer to the example given in our above response.
> > > * Change intro lines 45-46 to “Our primary motivation for using transcoders is **to enable input-invariant feature-level circuit analysis through MLP sublayers,** which allows us to understand and interpret the general behavior of circuits involving MLP sublayers.”
> > > * Move Section 4.1 (circuit analysis with transcoders) to come directly after Section 3.1 (transcoder architecture and training), in order to bring circuit analysis closer to the forefront of the paper. Then, move Section 3.2+3.3 (comparison with SAEs) and Section 4.2+4.3 (blind case studies and greater-than case study) into a “Comparison with SAEs section” (Section 4) and a “Circuit analysis case studies” section (Section 5).
> > > * This would also mean changing the order in which we present our contributions at the end of our introduction: we would be listing our discussion of circuit analysis as our first contribution, further foregrounding this aspect of our work.
> > > * At the beginning of the new Section 3.2 (circuit analysis with transcoders), include a brief discussion of why SAEs fail to provide input-invariant attributions through MLP sublayers. (This will consist of the “input-times-gradient” discussion in our response above.)
> > > * Move the conclusion of our discussion of circuit analysis through attention heads from Appendix D.3 (Eqn. 20) to the new Section 3.2.
> > >
> > > For reference, here is what the outline of the sections of our updated paper will be:
> > > 1. Intro
> > > 2. Preliminaries
> > > 3. Transcoders
> > >     1. Architecture
> > >     2. Circuit analysis (with new paragraph at beginning on drawbacks of circuit analysis with SAEs)
> > > 4. Comparison with SAEs
> > >     1. Quantitative SAE comparison
> > >     2. Qualitative SAE comparison
> > > 5. Circuit analysis case studies
> > >     1. Blind case studies
> > >     2. “Greater-than” circuit
> > > 6. Related work
> > > 7. Conclusion
> > >
> > > We are confident we can make these modifications without exceeding the camera-ready page limit. We already appreciate the valuable feedback that you have given us in the course of this conversation, and if you have any other suggestions for edits that would help to clarify things, then we would be more than happy to consider them.
> > >
> > > Again, thank you for your time and your comments.

---

### Author Rebuttal · Authors · 2024-08-07

**Summary:** We were happy to see our reviewers recognize transcoders’ importance for mechanistic interpretability and appreciate our input-invariant circuit analysis and blind case studies. We found that we might not have adequately conveyed the main goal of our work: to use transcoders to perform input-invariant circuit analysis which is impossible with SAEs. We have addressed this in our responses, and are reorganizing our paper to fix this.

We would like to thank the reviewers for the time they spent reading our paper and offering useful feedback. We were particularly glad to read Reviewer GwPg’s belief that “transcoders are maybe the most important idea for a new interpretability technique to investigate at the moment,” and their interest in how we “showcase using the technique to reverse engineer a circuit in section 4.3.” Indeed, we were happy to see that reviewers appreciated our blind case study procedure, with reviewer xRyp stating that “[t]he feature case studies are a valuable contribution to the literature” and reviewer Q9a1 stating that “[t]he blind evaluation protocol was really good to see---I appreciated authors' efforts in stress testing the limits of their approach (this is how good science should go :)).” We also were happy to see reviewers appreciate the utility of our input-invariant attribution decomposition, with reviewer sL5K stating that “circuit analysis presented in the paper is novel in the sense that it disentangles input-invariant information from input-dependent information,” and reviewer xRyp commenting that “[d]eriving the decomposition into input dependent / invariant features is novel and addresses a significant challenge of interpretability. On its own this warrants future exploration.”

With regard to reviewers’ concerns, the primary theme that we noticed across most reviews was a slight misunderstanding regarding our main purpose in writing this paper, which we feel might be due to a lack of clarity on our part. Concretely, **our main goal in our research was to develop a method for utilizing transcoders in weights-based, input-invariant circuit analysis.** In contrast, our experiments comparing transcoders to SAEs were primarily intended to corroborate that transcoders are on par with SAEs in interpretability and fidelity, in order for transcoders to be used in circuit analysis without incurring any penalties relative to SAEs. We believe that this misunderstanding may underlie certain reviewers’ emphasis on the experiments in Section 3.2 directly comparing SAEs and transcoders according to standard SAE metrics.

With this in mind, we encourage all reviewers to particularly assess transcoders with an eye towards the benefits that they bring to input-invariant circuit analysis. This is especially important because, as we explained in our response to xRyp, this sort of circuit analysis cannot be achieved with standard SAEs. We will update our paper to include a mathematical demonstration of why SAEs are insufficient. We will also restructure the ordering of sections in our paper by their importance to our main goal; concretely, this means putting our introduction to circuit analysis with transcoders before our section comparing them with SAEs.

Besides this, reviewers had some smaller separate questions, all of which we believe we have adequately addressed in our individual rebuttals.
Once again, we thank all of the reviewers, and the Area Chairs, for their time and valuable comments.

---

### Decision · Program_Chairs · 2024-09-25

**Decision:**

Accept (poster)

**Comment:**

The submission investigates transcoders, an encoder-decoder architecture, to imitate the input-output relationships in MLP layers in language models. Due to the linearity combination of MLP and attention blocks in the "residual stream", the contribution from a transcoder feature at a lower layer to the transcoder feature at a higher layer can be decomposed into input-dependent and input-invariant components. A similar analysis can be done for the attention heads. These can be combined with greedy searches to find sparse computation graphs that explain a specific transcoder feature.

The reviewers are generally enthusiastic about the submission: (i) the idea seems promising and could spur further work and (ii) the experiments are clear and clarifications during rebuttal were useful. The authors are encouraged to add some of these (e.g. reshuffling sections) to the camera-ready version.